# SPATIAL-SPECTRAL BINARIZED NEURAL NETWORK FOR PANCHROMATIC AND MULTI-SPECTRAL IMAGES FUSION

## ABSTRACT

Remote sensing pansharpening aims to reconstruct spatial-spectral properties during the fusion of panchromatic (PAN) images and low-resolution multi-spectral (LR-MS) images, finally generating the high-resolution multi-spectral (HR-MS) images. Although deep learning-based models have achieved excellent performance, they often come with high computational complexity, which hinder their applications on resource-limited devices. In this paper, we explore the feasibility of applying the binary neural network (BNN) to pan-sharpening. Nevertheless, there are two main issues with binarizing pan-sharpening models: (*i*) the binarization will cause serious spectral distortion due to the inconsistent spectral distribution of the PAN/LR-MS images; (*ii*) the common binary convolution kernel is difficult to adapt to the multi-scale and anisotropic spatial features of remote sensing objects, resulting in serious degradation of contours. To address the above issues, we design the customized spatial-spectral binarized convolution (S2B-Conv), which is composed of the Spectral-Redistribution Mechanism (SRM) and Gabor Spatial Feature Amplifier (GSFA). Specifically, SRM employs an affine transformation, generating its scaling and bias parameters through a dynamic learning process. GSFA, which randomly selects different frequencies and angles within a preset range, enables to better handle multi-scale and-directional spatial features. A series of S2B-Conv form a brand-new binary network for pan-sharpening, dubbed as S2BNet. Extensive quantitative and qualitative experiments have shown our high-efficiency binarized pan-sharpening method can attain a promising performance.

## 1 INTRODUCTION

With the increasing demand of earth observation and monitoring, existing optical satellites (e.g., GaoFen-2, QuickBird) can simultaneously record bundled low-resolution multispectral (LR-MS) and high-resolution panchromatic (PAN) images from the same scene. Due to physical limitations of existing satellite sensors, the recorded LR-MS image usually includes rich spectral property but relatively sparse spatial property, while their corresponding PAN image contains abundant spatial property but sparse spectral property. Since the remote sensing image with rich spatial-spectral properties, i.e., high-resolution multispectral (HR-MS) image, are crucial for practical applications Li et al. (2022); Han et al. (2024); Asokan & Anitha (2019), pansharpening technique, which could obtain the HR-MS image by reconstructing spatial-spectral properties during the fusion of the recorded PAN and LR-MS images, has been widely explored Xing et al. (2023).

Existing state-of-the-art (SOTA) pan-sharpening methods are based on deep learning. Convolutional neural network (CNN) and Transformer Wu et al. (2022); Tan et al. (2024); Li et al. (2023a) have been employed as powerful models to reconstruct spatial-spectral properties for target HR-MS images. For example, SRPPNN Cai & Huang (2020) performs the pan-sharpening learning using a deeper CNN network architecture implemented residual learning. Zhou *et al.* propose a series of Transformer-based algorithms push the performance boundary again Zhou et al. (2022b). Although superior performance is achieved, these CNN-/Transformer-based methods require powerful hardwares with abundant computing and memory resources, such as costing 439.38M in Hyper-Transformer Bandara & Patel (2022b). This motivates us to reduce the memory and computational

burden of pan-sharpening methods while preserving the performance as much as possible so that the algorithms can be deployed on resource-limited devices.

To improve the efficiency of deep neural networks, many network compression techniques are proposed, including network quantization Qin et al. (2020a), parameter pruning Wang & Fu (2022) and knowledge distillation Wang et al. (2022). Among these approaches, Binarized neural network (BNN) stands out as an extreme case of network quantization, which binarizes both weights and activations into only 1-bit. In particular, the foundation of BNNs lies in their pure logical computations, primarily XNOR and bit-count operations. This makes them highly energy-efficient, particularly for embedded devices Zhang et al. (2024). However, directly applying model binarization for pan-sharpening algorithms may face several challenges. **(i)** The density and distribution of the spectra within the LR-MS image are different, similarly, the spectral density and distribution between LR-MS and PAN images are also different (See in the Appendix). During the fusion of LR-MS and PAN images, equally binarizing the activations of different spectral channels may lead to severe distortion of spectral features. **(ii)** Remote sensing scenes are inherently multi-scale and anisotropic. Directly using the common binary convolutions, such as those used in BNNs, has two drawbacks: *(1)* the receptive field is limited, making it difficult to understand the global structure, resulting in blurred or disjointed contours; and *(2)* the convolution kernel is isotropic, which means it learns an "averaged" feature response during training, which can easily produce jagged effects or blurred edges.

Bearing the above challenges in mind, we propose a novel binarized method, namely Spatial-Spectral Binarized Neural Network (S2BNet) for efficient remote sensing pan-sharpening. Different from the recent binarized multi-spectral fusion network Hou et al. (2025), our work does not include complex computations like diffusion operation and attention mechanism that are difficult to implement on edge devices. **Firstly**, we design the basic unit, spatial-spectral binarized convolution (S2B-Conv), used in model binarization. S2B-Conv can adapt inconsistent spectral distribution and diverse spatial features before binarizing the activation. **Secondly**, the Spectral-Redistribution Mechanism (SRM) in S2B-Conv is employed to generate scaling and bias parameters in a adaptive learning manner, adjusting the density and distribution of spectral bands. **Thirdly**, we propose the Gabor Spatial Feature Amplifier (GSFA) in S2B-Conv, which captures multi-scale and-directional spatial features by randomly selecting different frequencies and angles within a preset range. **Finally**, we derive our S2BNet by using the proposed spatial-spectral binarized convolution to binarize the base model. As shown in Tab. 1, S2BNet achieves a substantial PSNR gain of over 2 dB against the E2FIF, and also outperforms the current best-performing method, BiSRNet, by more than 0.2 dB. In a nutshell, our contributions can be summarized as follows:

- We propose a novel BNN-based algorithm S2BNet, which is composed of several S2B-Conv units, for remote sensing pan-sharpening. In particular, the S2B-Conv could adapt complex spatial-spectral features before binarizing the model.
- The Spectral-Redistribution Mechanism (SRM) in S2B-Conv is introduced to adjust the distribution in spectral dimension through adaptive learning process.
- The Gabor Spatial Feature Amplifier (GSFA) in S2B-Conv is introduced to learn multi-scale and multi-directional spatial features by randomly selecting different frequencies and angles.
- Experiments on multiple remote sensing benchmark datasets show that the proposed S2BNet outperforms state-of-the-art binary neural networks.

## 2 RELATED WORK

### 2.1 PAN-SHARPENING

The classic pan-sharpening methods mainly consist of 3 categories: CS-based Gillespie et al. (1987), MRA-based Nunez et al. (1999), and VO-based methods Fasbender et al. (2008). Deep learning-based methods have significantly improved the performance of pan-sharpening tasks in recent times. As a groundbreaking study, PNN Masi et al. (2016) first introduced CNN-based methods into pan-sharpening tasks. PANNet Yang et al. (2017) leverages residual connections and high-frequency filtering techniques within a CNN framework. Moreover, HyperTransformer Bandara & Patel (2022a) was one of the early attempts to introduce Transformer-based methods into this field. Pan-

Former Zhou et al. (2022a) utilizes the Transformer framework to improve spatial resolution while maintaining spectral integrity.

Although deep learning models have achieved remarkable progress in pan-sharpening over traditional methods, they typically require powerful hardware with considerable memory and computational capacity, that hinders their deployment on resource-limited satellites. In this study, we explore the potential of using binarized neural networks for pan-sharpening to make the model more lightweight.

## 2.2 BINARIZED NEURAL NETWORK

Binary Neural Networks (BNNs) have emerged as a transformative paradigm for deploying deep learning models on resource-constrained devices, achieving drastic reductions in memory footprint and computational costs by representing weights and activations with -bit values Zhang et al. (2024). However, the extreme quantization introduces two fundamental and well-established challenges: the quantization error (severe information loss due to binarization) and the gradient error (mismatch in backward propagation due to the non-differentiable sign function) Hubara et al. (2016b); Liu et al. (2018).

The development of BNNs has been driven by systematic efforts to mitigate these core challenges. The pioneering work, BNN Hubara et al. (2016a), introduced the sign function and the Straight-Through Estimator (STE) Hubara et al. (2016b)as foundational solutions. Subsequent research evolved along clear technical strands: To combat quantization error, XNOR-Net Rastegari et al. (2016) introduced learnable scaling factors for weights and activations to better approximate full-precision operations. Recently, the exploration of BNNs has extended from high-level recognition tasks to low-level vision tasks like image super-resolution and enhancement, which demand high-fidelity pixel-wise reconstruction. For instance, Xin et al. Xin et al. (2020) designed a bit-accumulation mechanism to approximate full-precision convolution for super-resolution. Zhang et al. Zhang et al. (2024) tackled low-light video enhancement with BNNs, addressing challenges in temporal fusion and closing the gap with full-precision models. These works demonstrate the potential yet underscore the significant difficulty of applying BNNs to tasks requiring precise spatial and radiometric preservation.

The resource-constrained nature of satellite and edge devices makes model compression imperative for remote sensing applications. However, despite the progress in general BNNs and their nascent application in low-level vision, there remains a pronounced and unexplored gap: the adaptation of BNNs for high-fidelity remote sensing image fusion, particularly pansharpening. This task imposes unique dual requirements of spectral fidelity and spatial detail preservation that are acutely at odds with the inherent information bottleneck of 1-bit representation. The absence of dedicated research in this area highlights both the challenge and the opportunity that our work, S2BNet, seeks to address.

## 3 METHOD

In this section, we present the overall architecture of our network for pan-sharpening. Then, we elaborate on the details of the spatial-spectral binarized convolution, which mainly consists of the Spectral-Redistribution Mechanism and Gabor Spatial Feature Amplifier.

## 3.1 OVERALL ARCHITECTURE

Binary networks have achieved significant success in the fields of hyperspectral image reconstruction and low-light raw video enhancement. However, few studies have attempted to apply them to the task of pan-sharpening. Building on the existing research in low-level vision tasks and considering the characteristics of pan-sharpening, we design a network that focuses on both spatial and spectral attributes, as the reconstruction quality of these attributes greatly affects the outcome of pan-sharpening.

The general architecture of our S2BNet network is shown in Fig. 1. Inspired by Cai et al. (2023), we use a U-shaped structure for the base model. First, we upsample the LR-MS images and concatenate them with the Pan images, and feed the output into the full-precision convolutional layer to generate

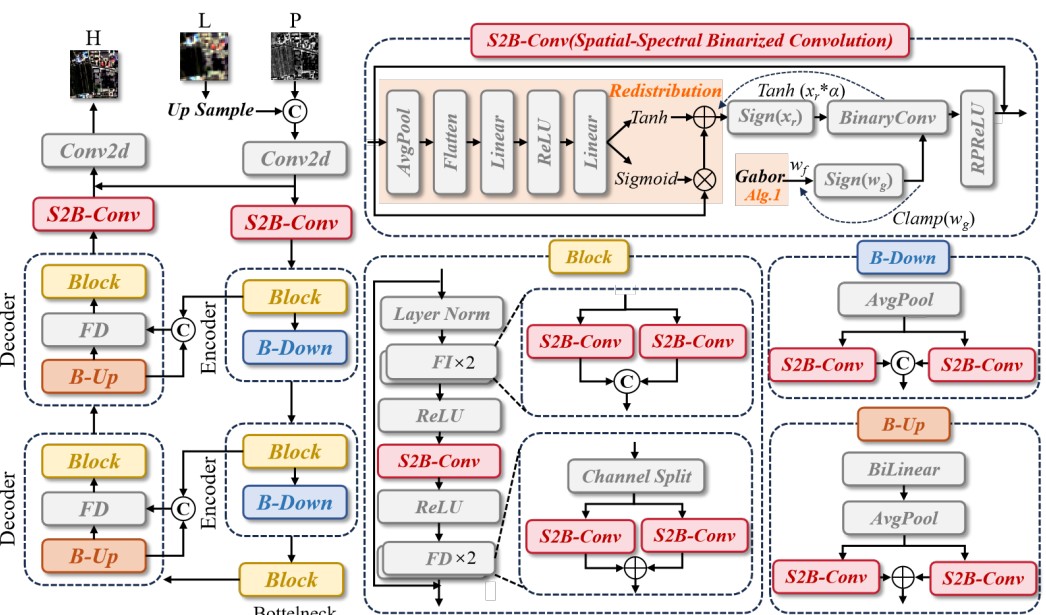

Figure 1: The overall frame of our method. Our model primarily consists of the novel convolution (S2B-Conv), which incorporates two core modules: the Spectral-Redistribution Mechanism (denoted as "Redistribution") and the Gabor Spatial Feature Amplifier (denoted as "Gabor"). The specific implementation of Gabor is detailed in Algorithm 1.

a shallow feature $X_s \in \mathbb{R}^{H \times W \times C}$ and subsequently into the S2B-Conv block. Here, $H$, $W$, and $C$ denote the height, width, and the number of channels. Then it undergoes two encoders, a bottleneck, and two decoders. To alleviate the information loss during processing, skip connections between the encoders and the decoders are employed. After that, the S2B-Conv block maps the features to generate the deep feature $X_d \in \mathbb{R}^{H \times W \times C}$. Finally, the sum of $X_s$ and $X_d$ is fed into another full-precision convolutional layer to produce the reconstructed HR-MS images. Specifically, each encoder consists of a basic block followed by a B-Down block, while each decoder is composed of a B-Up block, a Fusion Decrease block, and a basic block. Furthermore, the basic block serves as the bottleneck. It is worth noting that the entire network only employs two full-precision convolutions, while all other convolutions are binary convolutions.

As illustrated in Fig. 1, the Fusion Increase and Fusion Decrease modules both use binarized convolutional layers with a kernel size of 1 to aggregate the feature maps and modify the channels. The upsample module employs bilinear interpolation followed by binarized convolutional layers with a kernel size of 3 to upscale the feature maps and halve the channels, while the downsample module uses binarized convolutional layers with a kernel size of 3 to downscale the feature maps and double the channels. Moreover, the structure of the basic block is also presented in Fig. 1. It consists of Layer Norm, Fusion Increase block, S2B-Conv block, Fusion Decrease block, and ReLU activation function. The S2B-Conv block will be thoroughly discussed in subsequent chapters.

## 3.2 SPATIAL-SPECTRAL BINARIZED CONVOLUTION

As shown in the Fig. 1, we first adjust the spectral distribution of the full-precision activations $X_f \in \mathbb{R}^{H \times W \times C}$ using the Spectral-Redistribution Mechanism to get $X_r \in \mathbb{R}^{H \times W \times C}$ (which will be introduced in Section 3.2.1), and generate weights $W_f$ using the Gabor Spatial Feature Amplifier (which will be introduced in Section 3.2.2). Then, referring to Zhang et al. (2024); Cai et al. (2023), we binarize both $X_r$ and $W_f$, and fuse these two features using an XOR operation, as detailed subsequently.

The activation $X_r$ is binarized by a sign function in 1-bit values, producing $X_b \in \mathbb{R}^{H \times W \times C}$, where each element has a value of +1 or -1. The binarization procedure can be formulated as:

$$X_b = \text{Sign}(X_r) = \begin{cases} +1, & X_r > 0 \\ -1, & X_r \leq 0. \end{cases} \tag{1}$$

Since the Sign function is non-differentiable, we use a scalable hyperbolic tangent function to approximate the Sign function in the backpropagation as:

$$x_b = \text{Tanh}(\alpha x_r) = \frac{e^{\alpha x_r} - e^{-\alpha x_r}}{e^{\alpha x_r} + e^{-\alpha x_r}}, \tag{2}$$

where $\alpha \in \mathbb{R}^+$ is a learnable parameter that adaptively adjusts the distance between Tanh and Sign.

For weight $W_f$, we also utilize the sign function to generate the binarized weight $W_b$. And we adopt a piecewise linear function, Clip, during the backpropagation as:

$$W_b = \text{Clip}(W_f) = \begin{cases} +1, & W_f \geq 1 \\ x, & -1 < W_f < 1 \\ -1, & W_f \leq -1 \end{cases} \tag{3}$$

Subsequently, the computationally intensive floating-point matrix multiplication operations in full-precision convolution can be substituted with pure logical XNOR and bit-count operations as:

$$X_b \otimes W_b = \text{bitcount}(\text{XNOR}(X_b, W_b)). \tag{4}$$

Following Zhang et al. (2024), we multiply the mean absolute value of the 32-bit weight value $W_f$ to narrow down the difference between binarized and full-precision weights, i.e.,

$$S_i = \frac{\|W_f^i\|_1}{C \times K \times K}, \quad i = 1 \ldots C_{\text{out}}, \tag{5}$$

$$Y = (X_b \otimes W_b) \odot S, \tag{6}$$

where $S \in \mathbb{R}^{C_{\text{out}}}$ is the scaling factor and $Y \in \mathbb{R}^{H \times W \times C}$ denotes the output activation rescaled with the factors from weights.

To mitigate the loss of information from binarization, we introduce a residual connection that sums $X_f$ and $Y$. However, due to the significant difference in their value ranges, combining them directly using an identity mapping might obscure the information in $Y$. To resolve this issue, we first apply the $Y$ to RPReLU Liu et al. (2018), to adjust its value range before the summation. This step can be represented as:

$$X_o = X_f + \text{RPReLU}(Y), \tag{7}$$

where $X_o \in \mathbb{R}^{H \times W \times C}$ is the output feature and RPReLU is formulated as:

$$\text{RPReLU}(Y) = \begin{cases} Y_i - \gamma_i + \zeta_i, & Y_i > \zeta_i \\ \beta_i(Y_i - \gamma_i) + \zeta_i, & Y_i \leq \zeta_i, \end{cases} \tag{8}$$

where $\gamma_i, \zeta_i, \beta_i \in \mathbb{R}^{C_{\text{out}}}$ are learnable parameters.

### 3.2.1 THE SPECTRAL-REDISTRIBUTION MECHANISM

Upon analyzing the LR-MS images, we observe that the distribution and density of spectral information differ significantly across various bands Cai et al. (2023). (The visualization analysis of the spectral distribution is detailed in the Appendix.) The spectral information of Pan images and LR-MS images also has significant differences. This inconsistency in spectral information can negatively impact the reconstruction of spectral features, leading to potential inaccuracies in the final output.

Existing studies attempt to adjust the distribution of spectral information using affine transformations. However, they predominantly utilize a randomly initialized approach for scaling and biasing, which does not adapt to the input data. These methods lack the ability to dynamically adjust parameters based on input variations, potentially leading to suboptimal performance across diverse datasets.

To address this limitation, we introduce a Spectral-Redistribution Mechanism that generates scaling and bias parameters through a data-driven mechanism. This approach not only enhances adaptability to varying input characteristics but also enhances feature reconstruction accuracy by optimizing the redistribution process through learning global features.

As depicted in Fig. 1, first, the input feature $X_f$ undergoes global average pooling and is then flattened to obtain the global feature $X_g \in \mathbb{R}^{H \times W \times C}$. Subsequently, two fully connected layers along with a ReLU activation function are employed to generate the scaling factor $k$ and the bias $b$. Specifically, the number of channels of feature $X_g$ is first mapped from $C$ to $C/r$, and after passing through the ReLU, it is then mapped from $C/r$ to $2C$. Following this, the resulting feature is split along the channel dimension into two separate features, each with $C$ channels. Here $C$ and $r$ denote the number of channels and scaling factor, respectively. After that, we employ the sigmoid activation function to constrain the scaling factor $k$ within the range of 0 to 1, obtaining $k'$, and utilize the tanh function to limit the bias $b$ between -1 and 1, resulting in $b'$. Ultimately, the redistributed features are acquired through an affine transformation. The specific formulas are as follows:

$$k' = \sigma(k) = \frac{1}{1 + e^{-k}}, \tag{9}$$

$$b' = \tanh(b) = \frac{e^b - e^{-b}}{e^b + e^{-b}}, \tag{10}$$

$$X_r = X_f \cdot k' + b'. \tag{11}$$

### 3.2.2 GABOR SPATIAL FEATURE AMPLIFIER

The spectral characteristics of parking lots and open spaces, as well as bare land and leveled construction sites, are very similar and need to be distinguished by spatial features. However, their spatial shapes and sizes are also similar, and the resolution of the LR-MS images is too low. Therefore, a method that can extract spatial textures more accurately is needed.

To address this challenge, we propose the Gabor Spatial Feature Amplifier, which initializes the weights of convolutional layers using Gabor kernels. Gabor kernels, by randomly selecting different frequencies and angles within a preset range, enable subsequent convolutional layers to better handle multi-scale and multi-directional features, which is conducive to the learning of spatial features.

Although Gabor kernels are widely used in image processing to simulate the perception of edges and textures by the human visual system, existing research has not yet explored the application of Gabor kernels in binary convolutional networks.

First, we define two lists: one containing different frequency values and another containing different angle values. For each output channel, we randomly select a frequency and an angle from these lists to generate the Gabor kernel. The Gabor kernel is generated based on the following formula:

$$g(x, y; \lambda, \theta, \psi, \sigma, \gamma) = \exp\left(-\frac{x'^2 + \gamma^2 y'^2}{2\sigma^2}\right) \exp\left(i\left(2\pi\frac{x'}{\lambda} + \psi\right)\right), \tag{12}$$

where

$$\begin{cases} x' = x\cos\theta + y\sin\theta \\ y' = -x\sin\theta + y\cos\theta. \end{cases} \tag{13}$$

However, here we only use the real part, that is,

$$g(x, y; \lambda, \theta, \psi, \sigma, \gamma) = \exp\left(-\frac{x'^2 + \gamma^2 y'^2}{2\sigma^2}\right) \cos\left(2\pi\frac{x'}{\lambda} + \psi\right). \tag{14}$$

The parameters of the Gabor function each play a distinct role in shaping the characteristics of the filter. The parameter $\lambda$ signifies the wavelength of the sinusoidal component within the Gabor function. $\theta$ indicates the orientation or direction of the filter. The phase offset of the sinusoidal function is denoted by $\psi$. The parameter $\sigma$ represents the standard deviation of the Gaussian envelope, and $\gamma$ is the aspect ratio that influences the ellipticity of the Gaussian function. Since each output channel selects different parameters, the $C_{out}$ output channels will have $C_{out}$ different Gabor kernels, thus enabling the learning of spatial features at different directions and scales. Because the generated Gabor kernels are intended to replace the weights in the convolutional layer, we need to normalize

them so that their mean and variance are similar to the default weights of the convolutional layer. This ensures that the Gabor kernels are compatible with the initialization scheme typically used in convolutional neural networks. For the detailed algorithm, please refer to Algorithm. 1. For hyper-parameter selection, please refer to Sectiton 4.3.

---

**Algorithm 1** Gabor Spatial Feature Amplifier

---

**Require:** Number of input channels $in\_chn$, number of output channels $out\_chn$, number of groups $groups$, kernel size $kernel\_size$
**Ensure:** Gabor kernel $weight$
1: **while** not converged **do**
2:      $n\_in \leftarrow in\_chn \times kernel\_size \times kernel\_size$
3:      Initialize frequency array $freqs$ with $4$ elements
4:      Initialize angle array $thetas$ with $16$ elements
5:      $weight \leftarrow \text{empty}(out\_chn, \frac{in\_chn}{groups}, kernel\_size, kernel\_size, \text{dtype=np.float32})$
6:      **for** $oc \leftarrow 1$ to $out\_chn$ **do**             ▷ For each output channel, we generate a kernel.
7:         $\lambda \leftarrow$ random choice from $freqs$
8:         $\theta \leftarrow$ random choice from $thetas$
9:         $\gamma \leftarrow$ Set fixed parameter
10:        $\sigma \leftarrow$ Set fixed parameter based on $n_{in}$
11:        **for** $x \leftarrow 0$ to $kernel\_size - 1$ **do**
12:           **for** $y \leftarrow 0$ to $kernel\_size - 1$ **do**
13:             $x' \leftarrow x \cos(\theta) + y \sin(\theta)$
14:             $y' \leftarrow -x \sin(\theta) + y \cos(\theta)$
15:             $kernel \leftarrow \exp\left(-\frac{x'^2 + \gamma^2 y'^2}{2\sigma^2}\right) \cos\left(2\pi \frac{x'}{\lambda} + \psi\right)$
16:           **end for**
17:        **end for**
18:        $kernel \leftarrow kernel - \text{mean}(kernel)$
19:        $scale \leftarrow$ Set fixed parameter
20:        $kernel \leftarrow kernel \times scale$
21:        $weight[oc] \leftarrow kernel$
22:      **end for**
23:      **return** $weight$ as torch tensor
24: **end while**

---

## 3.3 LOSS FUNCTION

We use $L_1$ loss as the reconstruction loss:

$$\mathcal{L}_{rec} = \|G - H\|_1, \tag{15}$$

where $G$ is the ground truth image, and $H$ denotes the reconstructed image.

## 4 EXPERIMENTS

In this section, we evaluate our S2BNet on three pan-sharpening datasets. We also conduct extensive experiments to analyze our proposed model.

### 4.1 EXPERIMENTAL SETTINGS

**Datasets.** For the pan-sharpening, 4-band WorldView-2, 4-band GaoFen-2 and 8-band WorldView-3 datasets are adopted for experimental analysis. Due to the unavailability of ground-truth (GT) images for training, following Wald's protocol Wald et al. (1997). We employ downsampling operations to produce corresponding dataset for each satellite sensor, as shown in the Appendix, we present the detailed information of traning dataset and testing dataset in the experiment.

**Training Details.** Moreover, the proposed network is built with PyTorch and trained on four NVIDIA RTX $A$5000 GPUs. Training is conducted with a batch size of 16, and the Adam optimizer

Table 1: Quantitative comparison of our S2BNet with our full-precision and binary methods on the GaoFen-2 dataset.

| Category | Method | Params (K) | Flops (G) | Reduced-Resolution | | | | | Full-Resolution | | |
|---|---|---|---|---|---|---|---|---|---|---|---|
| | | | | PSNR↑ | SSIM↑ | $Q_4$↑ | SAM↓ | ERGAS↓ | $D_\lambda$↓ | $D_s$↓ | QNR↑ |
| Full | DMFNet Yuan et al. (2019) | 1631.924 | 145.811 | 43.1223 | 0.9734 | 0.8557 | 0.0322 | 1.5660 | 0.0678 | 0.1134 | 0.8265 |
| | PANFormer Zhou et al. (2022a) | 1530.300 | 12.002 | 44.8501 | 0.9805 | 0.8865 | 0.0271 | 1.3337 | 0.0670 | 0.1806 | 0.7639 |
| | MutInf Zhou et al. (2022c) | 185.496 | 9.986 | 44.8306 | 0.9800 | 0.8836 | 0.0277 | 1.3394 | 0.0755 | 0.1762 | 0.7613 |
| | FAMENet Xuanhua et al. (2024) | 1244.228 | 39.272 | 45.6617 | 0.9837 | 0.8966 | 0.0248 | 1.2142 | 0.0697 | 0.1800 | 0.7622 |
| | CANNet Duan et al. (2024) | 785.118 | 3.237 | 45.2222 | 0.9816 | 0.8869 | 0.0267 | 1.2748 | 0.0654 | 0.1969 | 0.7498 |
| | UGCC Zeng et al. (2025) | 233.233 | 307.495 | 40.5620 | 0.9586 | 0.7803 | 0.0448 | 2.1717 | 0.0893 | 0.0608 | 0.8564 |
| | MSCSCFormer Ye et al. (2024) | 1950.686 | 582.435 | 42.6871 | 0.9696 | 0.8350 | 0.0360 | 1.7149 | 0.0692 | 0.0973 | 0.8403 |
| Binary | BNN Hubara et al. (2016a) | 17.760 | 1.938 | 34.4931 | 0.8802 | 0.4701 | 0.0860 | 4.1576 | 0.0985 | 0.0871 | 0.8243 |
| | LCRBNN Shang et al. (2022) | 17.760 | 1.938 | 32.2058 | 0.8165 | 0.2852 | 0.0952 | 5.4921 | 0.1043 | 0.1621 | 0.7503 |
| | BiSRNet Cai et al. (2023) | 26.459 | 0.664 | 43.3446 | 0.9727 | 0.8424 | 0.0327 | 1.5748 | 0.0669 | 0.0781 | 0.8604 |
| | BBCU Xia et al. (2022) | 23.520 | 0.664 | 43.0233 | 0.9712 | 0.8361 | 0.0335 | 1.6449 | 0.0676 | 0.0900 | 0.8484 |
| | FABNet Jiang et al. (2023) | 59.296 | 0.664 | 43.0144 | 0.9709 | 0.8370 | 0.0342 | 1.6477 | 0.0676 | 0.0867 | 0.8515 |
| | IRNet Qin et al. (2020b) | 17.760 | 1.938 | 42.8422 | 0.9714 | 0.8394 | 0.0338 | 1.6636 | 0.0692 | 0.1095 | 0.8285 |
| | E2FIF Song et al. (2023) | 20.640 | 0.664 | 41.0767 | 0.9544 | 0.7992 | 0.0402 | 2.0694 | 0.0740 | 0.0909 | 0.8422 |
| | **S2BNet** | 75.407 | 0.920 | **43.6495** | **0.9746** | **0.8503** | **0.0313** | **1.5206** | **0.0641** | **0.0710** | **0.8699** |

is adopted, setting $\beta_1 = 0.9$ and $\beta_2 = 0.999$ for its momentum coefficients. The initial learning rate is set to $1.5 \times 10^{-3}$ and is multiplied by $0.85$ every 100 epochs. The whole training procedure runs for 1500 epochs before stopping.

**Metrics.** Following previous studies on pan-sharpening, five image quality assessment metrics Yang et al. (2023) are employed for evaluation on reduced-resolution images, including spectral angle mapper (SAM), dimensionless global error in synthesis (ERGAS), the structural similarity (SSIM), the peak signal-to-noise ratio (PSNR), and the Q-index. Since there are not GT images available for the full-resolution dataset, we use three non-reference metrics to evaluate the performance of the model: Spectral Distortion Index ($D_\lambda$), Spatial Distortion Index ($D_s$), and No-Reference Quality (QNR) Alparone et al. (2008).

**Efficiency Evaluation.** Following previous BNN work Cai et al. (2023), the FLOPs per second for BNN is determined by $\text{Flops}^{\text{b}} = \text{Flops}^{\text{f}}/64$. The overall FLOPs is $\text{Flops}^{\text{b}} + \text{Flops}^{\text{f}}$. The parameters of BNN are calculated as $\text{Params}^{\text{b}} = \text{Params}^{\text{f}}/32$. The total number of parameters is $\text{Params}^{\text{b}} + \text{Params}^{\text{f}}$.

### 4.2 COMPARE WITH STATE-OF-THE-ARTS

**Comparison Methods.** We compare our method with various full precision pan-sharpening networks including DMFNet Yuan et al. (2019), PANFormer Zhou et al. (2022a), MutInf Zhou et al. (2022c), FAMENet Xuanhua et al. (2024), CANNet Duan et al. (2024), UGCC Zeng et al. (2025) and MSCSCFormer Ye et al. (2024). We also compare our spatial-spectral binarized convolution with other binarization methods, including BNN Hubara et al. (2016a), LCRBNN Shang et al. (2022), BiSRNet Cai et al. (2023), BBCU Xia et al. (2022), FABNet Jiang et al. (2023), IRNet Qin et al. (2020b) and E2FIF Song et al. (2023).

**Quantitative Comparison.** As summarized in Tab. 1 and Tab. 2, our method delivers substantial performance improvements over other binary techniques across all metrics, showcasing superior spectral and spatial fidelity. Lower SAM values demonstrate its effective spectral preservation, while higher PSNR scores confirm its robust spatial detail retention. More importantly, our model surpasses some of the full-precision baselines, demonstrating its promising expressiveness while benefiting from the efficiency of model binarization.

We additionally assess our model's generalization capability in real-world scenarios by testing it on full-resolution data. As shown in Tab. 1 and Tab. 2, our model consistently achieves superior performance across all three datasets, yielding the optimal scores on most evaluation metrics.

The quantitative results on the WorldView-3 examples are shown in Appendix.

Table 2: Quantitative comparison of our S2BNet with our full-precision and binary methods on the WorldView-2 dataset.

| Category | Method | Params (K) | FLOPs (G) | Reduced-Resolution | | | | | Full-Resolution | | |
|---|---|---|---|---|---|---|---|---|---|---|---|
| | | | | PSNR↑ | SSIM↑ | $Q_4$↑ | SAM↓ | ERGAS↓ | $D_\lambda$↓ | $D_s$↓ | QNR↑ |
| Full | DMFNet Yuan et al. (2019) | 1631.924 | 145.811 | 41.0268 | 0.9716 | 0.8184 | 0.0252 | 1.0855 | 0.0644 | 0.0863 | 0.8557 |
| | PanFormer Zhou et al. (2022a) | 1530.300 | 12.002 | 41.3495 | 0.9731 | 0.8237 | 0.0242 | 1.0621 | 0.0628 | 0.0844 | 0.8590 |
| | MutInf Zhou et al. (2022c) | 185.496 | 9.986 | 41.9527 | 0.9760 | 0.8258 | 0.0227 | 1.0152 | 0.0622 | 0.0794 | 0.8643 |
| | FameNet Xuanhua et al. (2024) | 1244.228 | 39.272 | 42.0278 | 0.9768 | 0.8332 | 0.0222 | 0.9936 | 0.0624 | 0.0753 | 0.8678 |
| | CANNet Duan et al. (2024) | 785.118 | 3.237 | 41.5868 | 0.9737 | 0.8256 | 0.0237 | 1.0517 | 0.0612 | 0.0767 | 0.8679 |
| | UGCC Zeng et al. (2025) | 233.233 | 307.495 | 35.8744 | 0.9296 | 0.6827 | 0.0460 | 1.9253 | 0.0887 | 0.0872 | 0.8327 |
| | MSCSCFormer Ye et al. (2024) | 1950.686 | 582.435 | 40.5338 | 0.9691 | 0.7963 | 0.0270 | 1.2016 | 0.0654 | 0.0772 | 0.8634 |
| Binary | BNN Hubara et al. (2016a) | 17.760 | 1.938 | 32.9122 | 0.8706 | 0.4595 | 0.0679 | 3.1063 | 0.0970 | 0.1244 | 0.7916 |
| | LCRBNN Shang et al. (2022) | 17.760 | 1.938 | 29.3907 | 0.7781 | 0.1877 | 0.0886 | 4.9267 | 0.0968 | 0.2997 | 0.6323 |
| | BiSRNet Cai et al. (2023) | 26.459 | 0.664 | 40.5263 | 0.9680 | 0.8016 | 0.0270 | 1.1755 | 0.0631 | 0.0756 | 0.8670 |
| | BBCU Xia et al. (2022) | 23.520 | 0.664 | 40.4211 | 0.9675 | 0.7957 | 0.0272 | 1.1904 | 0.0627 | 0.0758 | 0.8671 |
| | FABNet Jiang et al. (2023) | 59.296 | 0.664 | 40.0005 | 0.9647 | 0.7898 | 0.0287 | 1.2297 | 0.0624 | **0.0748** | **0.8684** |
| | IRNet Qin et al. (2020b) | 17.760 | 1.938 | 39.2731 | 0.9615 | 0.7736 | 0.0311 | 1.3434 | 0.0687 | 0.0801 | 0.8577 |
| | E2FIF Song et al. (2023) | 20.640 | 0.664 | 37.1290 | 0.9407 | 0.7273 | 0.0412 | 1.6997 | 0.0741 | 0.0833 | 0.8497 |
| | **S2BNet** | 75.407 | 0.920 | **40.6059** | **0.9684** | **0.8035** | **0.0267** | **1.1742** | **0.0619** | 0.0760 | 0.8677 |

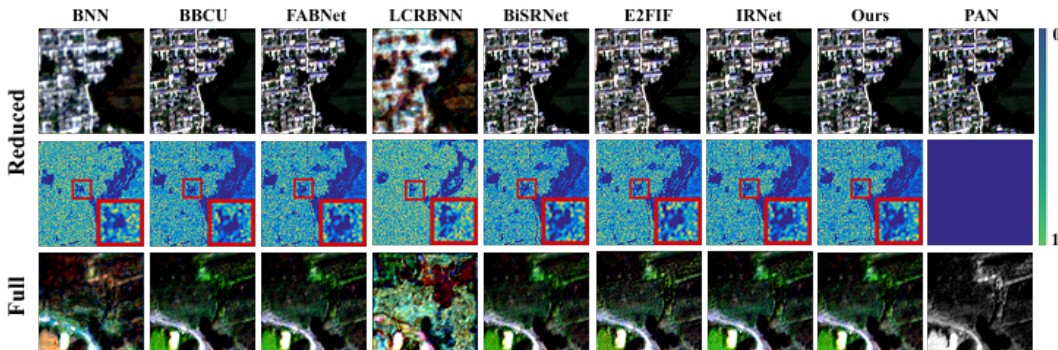

Figure 2: Visual comparison between our model and other binary methods on WV-2 example. The top two lines represent the reconstructed results and corresponding MAE maps of the reduced-resolution example, and the last line represents the reconstructed results of the full-resolution example.

**Visual Comparison.** The qualitative results on the WorldView-2 dataset are illustrated in Fig. 2. It can be observed that our model produces image with clear textures and visually pleasing spectra, and the corresponding MAE map exhibits minimal bright spots, indicating a high resemblance to the ground truth. In addition, the image generated by our S2BNet exhibits reduced aberrations and artifacts in full-resolution example. The visualization results on the GaoFen-2 example are shown in Appendix.

**Performance-Efficiency Comparison.** The performance-efficiency comparison between our method and other full-precision and binary benchmarks is also presented in Tab. 1. Compared with full-precision methods, our approach not only surpasses most methods in performance but also minimizes computational overhead. Compared to other binary models, our S2BNet achieves the highest PSNR score, with a marginally lower FLOPs than IRNet.

### 4.3 ABLATION STUDY

**Effect of the Spectral Redistribution Mechanism.** To verify the effect of Spectral-Redistribution Mechanism (SRM) in the S2B-Conv, Config (II) in Tab. 3 is obtained by removing Spectral Redistribution Mechanism, and uses random initial scaling and bias values. It could be reported in Tab. 3 that the model achieves 0.31 dB improvement when we exploit SRM.

Table 3: Comparison of different configurations.

| Config | GSFA | SRM | PSNR↑ | SSIM↑ | $Q_4$↑ | SAM↓ | ERGAS↓ | $D_\lambda$↓ | $D_s$↓ | QNR↑ |
|--------|------|-----|-------|-------|--------|------|--------|--------------|--------|------|
| (I) | ✗ | ✓ | 43.1074 | 0.9721 | 0.8358 | 0.0333 | 1.6159 | 0.0657 | 0.0896 | 0.8507 |
| (II) | ✓ | ✗ | 43.4300 | 0.9731 | 0.8411 | 0.0322 | 1.5651 | 0.0673 | 0.0977 | 0.8414 |
| Ours | ✓ | ✓ | **43.6495** | **0.9746** | **0.8503** | **0.0313** | **1.5206** | **0.0641** | **0.0710** | **0.8699** |

**Effect of the Gabor Spatial Feature Amplifier.** As shown in Tab. 3, the Config (I) is denoted, which replaces the Gabor Spatial Feature Amplifier (GSFA) with traditional convolutional weights. Our GSFA dramatically surpasses the common convolutional weights by $0.46$ dB.

**Binarizing Different Parts.** In the case of the hyperparameters $x = 5$ and $y = 16$ for GSFA, we binarize one part of the S2BNet while keeping the other parts full-precision to study the binarization benefit. The results are reported in Tab. 4. The base model (full precision) yields 42.2759 dB in PSNR while costing 13.884G OPs and 349.695K Params. We can find that (i) Binarizing the bottleneck reduces the Params the most (100.329K) with the smallest performance drop (only 0.16 dB). (ii) Binarizing the decoder achieves the largest OPs reduction (6.177G) while the performance degrades by a moderate margin (0.57 dB).

Table 4: Ablation study of binarizing different parts of the base model.

| | **Flops$^f$** | **Flops$^b$** | **Params$^f$** | **Params$^b$** | **Flops$^t$** | **Params$^t$** | PSNR↑ |
|--------|------|------|------|------|------|------|------|
| Encoder | 4.194 | 0.066 | 85.504 | 2.672 | 9.755 | 266.863 | 41.3088 |
| Bottelneck | 1.493 | 0.023 | 100.352 | 3.136 | 12.414 | 252.479 | 42.1118 |
| Decoder | 6.275 | 0.098 | 88.064 | 2.752 | 7.707 | 264.383 | 41.7051 |

## 5 CONCLUSION

In this paper, we propose a novel BNN-based method, S2BNet, for pan-sharpening. S2BNet is the compact and easy-to-deploy base model with simple computation operations. Specifically, we customize the basic unit S2B-Conv for model binarization. S2B-Conv utilizes the Spectral-Redistribution Mechanism, which can adaptively adjust the density and distribution of spectral features. Besides, S2B-Conv employs the Gabor Spatial Feature Amplifier to capture multi-scale and multi-directional spatial features. Comprehensive quantitative and qualitative experiments demonstrate that our S2BNet significantly outperforms SOTA BNNs and even achieves comparable performance with full-precision pan-sharpening algorithms. While these design is inspiring, their generalization ability in other network architectures (such as Transformer) is a valuable direction for future research.

## 6 REPRODUCIBILITY STATEMENT

To ensure the reproducibility of our research, we will make the core code publicly available in the supplementary materials.

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

## A    THE USE OF LLMS

We utilized GPT and Kimi to correct grammatical errors and translate some sentences.

## B    APPENDIX

**Spectral Visualization Analysis.** As shown in Fig. 3, we plot the spectral distribution using four different methods. The four subplots are described as follows:

Per-band Mean  Std (top-left): The mean (bar height) and standard deviation (error bars) of each band are displayed. Differences in bar height indicate uneven frequency response, while varying error-bar lengths reflect different intra-band fluctuation ranges, demonstrating spectral non-uniformity from a central-trend perspective.

Per-band Distribution (top-right): Box plots present the median, inter-quartile range, and outliers for every band. Noticeable shifts in box position and height reveal significant disparities in central values and dispersion across bands, further quantifying inconsistent frequency distributions.

Histogram Overlay (bottom-left): Overlaid histograms of all bands are plotted. Offset peaks and differing widths/shapes show that frequency-level distributions vary in range and form, illustrating spectral-response non-uniformity in terms of probability density.

Horizontal Profile (bottom-right): Pixel values along the central image row are plotted column-wise for each band. Divergent curve trends and amplitudes visually confirm spatially uneven frequency responses, verifying inconsistent relative brightness relationships among bands.

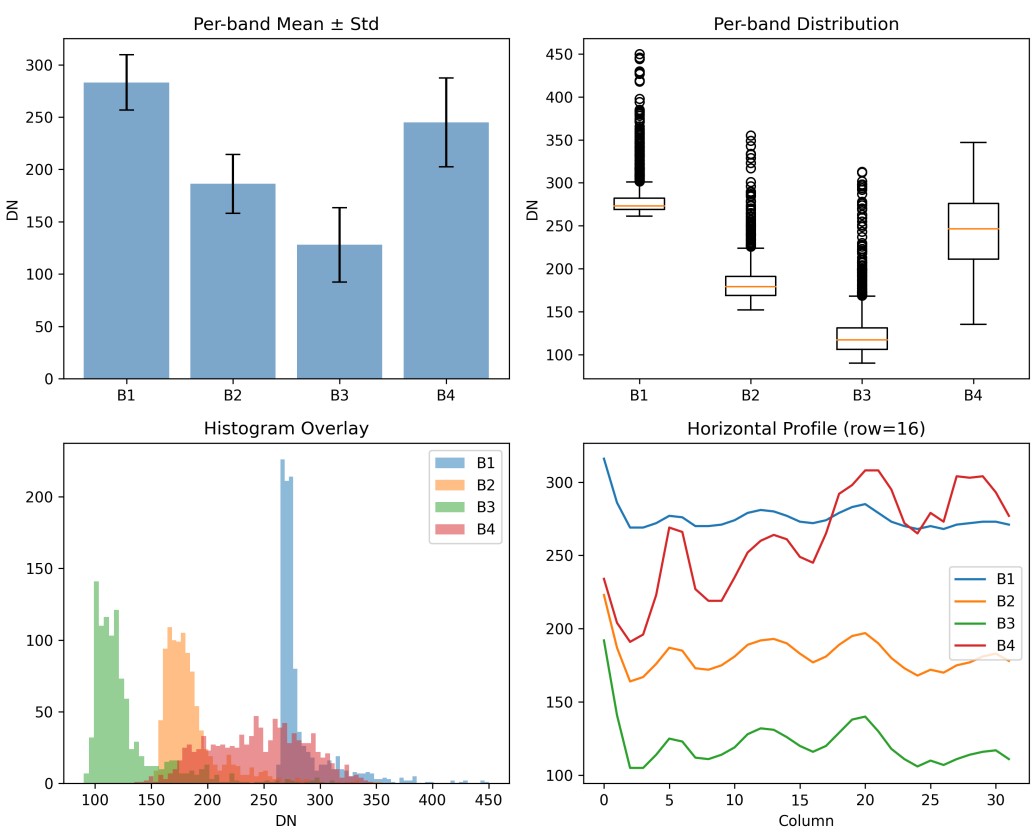

Figure 3: Spectral Distribution Plotted by Four Different Methods.

**Pan-sharpening Datasets.** We conduct experiments using the widely recognized WorldView-2, WorldView-3 and GaoFen-2 datasets Li et al. (2023b). The WorldView-2 dataset consists of in-

stances acquired by the sensor aboard the WorldView-2 satellite. This sensor captures data which covers wavelengths from $0.4$ to $1$ $\mu$m, with a spatial resolution of $1.24$ m. The WorldView-3 dataset consists of instances acquired by the sensor aboard the WorldView-3 satellite. This sensor captures data across eight multispectral bands and one panchromatic band, with spatial resolutions of $1.24$ meters for the multispectral bands and $0.31$ meters for the panchromatic band. The multispectral bands cover wavelengths from approximately $400$ to $1040$ nanometers. The images in the GaoFen-2 dataset are collected by the sensor onboard the GaoFen-2 satellite, which records data across four spectral bands within the wavelength range of $0.45 - 0.89$ $\mu$m. Additionally, this sensor provides a spatial resolution of $3.2m$. As shown in Tab. 5, we present the detailed information of training dataset and testing dataset in the experiment.

Table 5: The detailed dataset information (GaoFen-2, WorldView-2 and WorldView-3), followed by Ma et al. (2025).

| Sensor | bit depth | #Images | Scale | Training | Testing |
|---|---|---|---|---|---|
| WordView-2 | 11 | 1 | Reduced | **#Patches:** 1012
**PAN:** $128 \times 128 \times 1$
**LR-MS:** $32 \times 32 \times 4$
**Output:** $128 \times 128 \times 4$ | **#Patches:** 145
**PAN:** $128 \times 128 \times 1$
**LR-MS:** $32 \times 32 \times 4$
**Output:** $128 \times 128 \times 4$ |
| | | | Full | -
-
-
- | **#Patches:** 120
**PAN:** $128 \times 128 \times 1$
**LR-MS:** $32 \times 32 \times 4$
**Output:** $128 \times 128 \times 4$ |
| WorldView-3 | 11 | 1 | Reduced | **#Patches:** 910
PAN: $128 \times 128 \times 1$
**LR-MS:** $32 \times 32 \times 8$
**Output:** $128 \times 128 \times 8$ | **#Patches:** 144
PAN: $128 \times 128 \times 1$
**LR-MS:** $32 \times 32 \times 8$
**Output:** $128 \times 128 \times 8$ |
| | | | Full | -
-
-
- | **#Patches:** 120
**PAN:** $128 \times 128 \times 1$
**LR-MS:** $32 \times 32 \times 8$
**Output:** $128 \times 128 \times 8$ |
| GaoFen-2 | 11 | 1 | Reduced | **#Patches:** 1036
**PAN:** $128 \times 128 \times 1$
**LR-MS:** $32 \times 32 \times 4$
**Output:** $128 \times 128 \times 4$ | **#Patches:** 136
**PAN:** $128 \times 128 \times 1$
**LR-MS:** $32 \times 32 \times 4$
**Output:** $128 \times 128 \times 4$ |
| | | | Full | -
-
-
- | **#Patches:** 120
**PAN:** $128 \times 128 \times 1$
**LR-MS:** $32 \times 32 \times 4$
**Output:** $128 \times 128 \times 4$ |

**More experimental results.** To show the effectiveness of the proposed method, we also present the visualize the quantitative results and visual results of the proposed method on other pan-sharpening datasets in Tab. 2, Tab. 6,and Fig. 4.

Table 6: Quantitative comparison of our S2BNet with binary methods on the 8-band WorldView-3 dataset.

| Methods | Full-Resolution | | | | | Reduced-Resolution | | |
|---|---|---|---|---|---|---|---|---|
| | PSNR↑ | SSIM↑ | $Q_8$ ↑ | SAM↓ | ERGAS↓ | $D_\lambda$ ↓ | $D_S$ ↓ | QNR↑ |
| BNN Hubara et al. (2016a) | 23.8741 | 0.7228 | 0.7002 | 0.1109 | 6.7457 | 0.0423 | **0.0442** | 0.9154 |
| LCRBNN Shang et al. (2022) | 21.2906 | 0.486 | 0.4255 | 0.1467 | 8.9888 | 0.0384 | 0.0774 | 0.8871 |
| BiSRNet Cai et al. (2023) | 30.8967 | 0.9396 | 0.933 | 0.0691 | 3.0525 | 0.0151 | 0.0526 | 0.9331 |
| BBCU Xia et al. (2022) | 31.0004 | 0.9412 | 0.9351 | 0.0687 | 3.0178 | 0.0154 | 0.0537 | 0.9317 |
| FABNet Jiang et al. (2023) | 31.0697 | 0.9421 | 0.9358 | 0.0678 | 2.992 | 0.0155 | 0.0552 | 0.9301 |
| IRNet Qin et al. (2020b) | 30.1481 | 0.9274 | 0.9196 | 0.0759 | 3.3315 | 0.0191 | 0.0556 | 0.9263 |
| E2FIF Song et al. (2023) | 30.0244 | 0.9256 | 0.9182 | 0.075 | 3.3676 | 0.0218 | 0.0524 | 0.9268 |
| Ours | **31.2003** | **0.9441** | **0.9375** | **0.0661** | **2.962** | **0.014** | 0.0508 | **0.9359** |

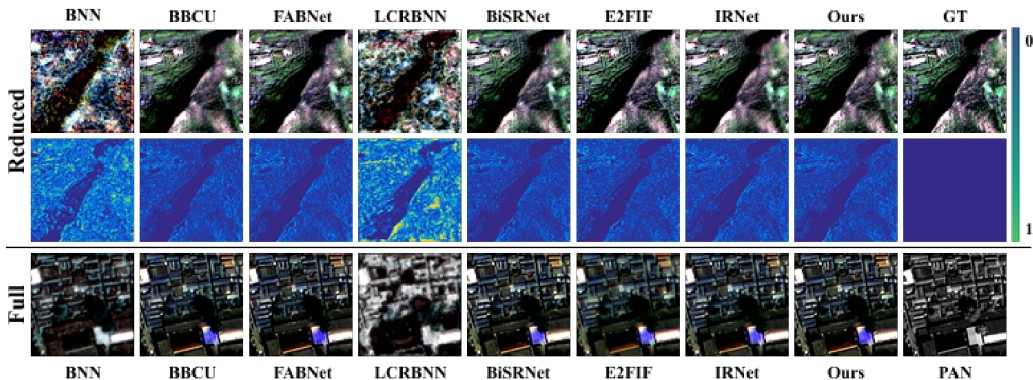

Figure 4: Visual comparison between our model and other binary methods on GF-2 example. The top two lines represent the reconstructed results and corresponding MAE maps of the reduced-resolution example, and the last line represents the reconstructed results of the full-resolution example.

**Difference between adaptive scaling/Gabor filters and Our SRM/GSFA**:

(1) SRM vs. SE: The designed SRM introduces an offset parameter (via Tanh), while SE only has a scaling parameter. This changes the modulation from purely multiplicative to affine, allowing for more flexible adjustment of the feature distribution.

(2) SRM vs. FiLM: The parameter generation of the designed SRM is adaptive (from the input features themselves), while FiLM depends on external conditions. Therefore, the designed SRM focuses more on internal feature recalibration, while FiLM focuses on cross-modal feature fusion. In summary, the designed SRM offers the flexible control capabilities in spectral processing, enabling fine-grained spectral optimization with input adaptation.

(3) GSFA vs. Gabor: GSFA fundamentally transcends the limitations of fixed parameters and linear convolution in classic Gabor filters by introducing learnable mechanisms and nonlinear hierarchies to generate higher-quality spatial features. Instead of using manually preset Gabor parameters, it adaptively optimizes the core properties of the filter, such as orientation and frequency. Simultaneously, the introduction of binarization operations (Sign, BinaryConv) and nonlinear activation functions (Tanh, RPReLU) forces the network to learn spatial features with less noise and greater robustness, achieving hierarchical nonlinear representations from simple edges to complex textures. Ultimately, the entire GSFA functions as an end-to-end component, enabling it to generate high-quality spatial features that are rich in detail, highly discriminative, and task-relevant.

Table 7: Experimental results using different $x$ and $y$ on the GaoFen-2 dataset

| $x$ | $y$ | Reduced-Resolution | | | | | Full-Resolution | | |
|---|---|---|---|---|---|---|---|---|---|
| | | PSNR↑ | SSIM↑ | $Q_4$↑ | SAM↓ | ERGAS↓ | $D_\lambda$↓ | $D_S$↓ | QNR↑ |
| 3 | 8 | 43.6362 | 0.9744 | 0.8486 | 0.0315 | 1.5254 | 0.0678 | 0.0786 | 0.8591 |
| 3 | 16 | 43.1619 | 0.9722 | 0.8416 | 0.0332 | 1.6055 | 0.0692 | 0.1034 | 0.834 |
| 3 | 32 | 43.1149 | 0.9715 | 0.8382 | 0.0331 | 1.6172 | 0.0663 | 0.1082 | 0.8325 |
| 5 | 8 | 43.1584 | 0.9717 | 0.8346 | 0.0332 | 1.6082 | 0.0658 | 0.0752 | 0.8643 |
| 5 | 16 | 43.5619 | 0.9745 | 0.8523 | 0.0317 | 1.5384 | 0.0655 | 0.0680 | 0.8714 |
| 5 | 32 | 43.4334 | 0.9735 | 0.8463 | 0.0323 | 1.5549 | 0.0667 | 0.1176 | 0.8229 |
| 7 | 8 | 43.4779 | 0.974 | 0.848 | 0.0323 | 1.5648 | 0.0664 | 0.1126 | 0.8283 |
| 7 | 16 | 43.7681 | 0.9752 | 0.8537 | 0.0312 | 1.5034 | 0.0651 | 0.1022 | 0.8395 |
| 7 | 32 | 43.6495 | 0.9746 | 0.8503 | 0.0313 | 1.5206 | 0.0641 | 0.0710 | 0.8699 |

**Hyperparameter selections of GSFA.** We use freqs $= \{\frac{\pi}{2} \cdot 2^{-(n-1)/2} \mid n \in \mathbb{N}, 1 \leq n < x\}$ to generate the frequency array mentioned in Algorithm 1, and use thetas $= \{\frac{k\pi}{y} \mid k \in \mathbb{Z}, 0 \leq k < y\}$ to generate the angle array. Tab. 7 presents the experimental results regarding the selection of hyperparameters $x$ and $y$.

