# OpenReview forum: "Spatial-Spectral Binarized Neural Network for Panchromatic and Multi-spectral Images Fusion"
_ICLR.cc/2026/Conference — Submitted to ICLR 2026_

### Official Review · Reviewer_HWxd · 2025-10-27

**Soundness:** 3
**Presentation:** 3
**Contribution:** 2
**Rating:** 4
**Confidence:** 3

**Summary:**

The paper tackles remote sensing pansharpening—fusing a high-resolution panchromatic (PAN) image with a low-resolution multispectral (LR-MS) image to produce a high-resolution multispectral (HR-MS) image—under tight compute and memory constraints. The authors propose S2BNet, a largely binarized U-shaped network for efficient pan-sharpening. Its core is a Spatial-Spectral Binarized Convolution (S2B-Conv) that combines with Spectral-Redistribution Mechanism (SRM) and Gabor Spatial Feature Amplifier (GSFA).
S2BNet places these modules within a U-shaped architecture (dual encoders, bottleneck, dual decoders with skip connections), using binary convolutions for most layers and a small number of full-precision layers. Experiments on GaoFen-2, WorldView-2, and QuickBird (4-band) with standard Wald protocol and metrics (e.g., PSNR, SSIM, SAM, ERGAS, Q-index, QNR and its components) show that S2BNet outperforms other binary approaches and is competitive with some full-precision baselines. Ablation studies indicate both SRM and GSFA contribute measurable gains, and binarizing different parts trades accuracy for parameter/OPs reduction. The authors claim code will be released.

**Strengths:**

- Practical and targeted adaptation of BNNs to pansharpening, addressing two concrete pain points (spectral mismatch and anisotropic/multi-scale spatial textures) via SRM and GSFA.
- Extensive experiments on three common satellites (GF-2, WV-2, QB), with strong baselines including recent CNN/Transformer and binary methods; both quantitative and qualitative results; ablations showing each component’s effect and binarization placement trade-offs.
- Architecture and modules are explained clearly; equations for SRM and Gabor filters help reproducibility.

**Weaknesses:**

Limited novelty: SRM is essentially a bounded affine calibration and GSFA leverages classical Gabor filtering with randomized parameterization. While these yield practical gains, the conceptual advance is incremental relative to recent pansharpening models introducing stronger spatial–spectral mechanisms, such as transformer-based fusion [1], content-adaptive non-local convolution [2], adaptive kernel-shape learning [3], and frequency-domain mixture-of-experts [4].

Generalization and robustness: No cross-sensor generalization (train on one sensor, test on another). In full-resolution evaluation, per-band spectral fidelity (e.g., band-wise SAM) and spectral index errors (e.g., NDVI) are not reported. There is no sensitivity analysis for GSFA parameter ranges nor an exploration of learnable Gabor parameters, despite recent emphasis on adaptivity in spatial kernels and frequency components [2][3][4].

[1] Zhou, H., Liu, Q., & Wang, Y. (2022). PanFormer: A transformer based model for pan-sharpening. 2022 IEEE International Conference on Multimedia and Expo (ICME). arXiv:2203.02916

[2] Duan, Y., Wu, X., Deng, H., & Deng, L.-J. (2024). Content-adaptive non-local convolution for remote sensing pansharpening. Proceedings of the IEEE/CVF Conference on Computer Vision and Pattern Recognition, 27738–27747.

[3] Wang, X., Zheng, Z., Shao, J., Duan, Y., & Deng, L.-J. (2025). Adaptive rectangular convolution for remote sensing pansharpening. Proceedings of the IEEE/CVF Conference on Computer Vision and Pattern Recognition, 17872–17881.

[4] He, X., Yan, K., Li, R., Xie, C., Zhang, J., & Zhou, M. (2024). Frequency-adaptive pan-sharpening with mixture of experts. Proceedings of the AAAI Conference on Artificial Intelligence, 38(3), 2121–2129. https://doi.org/10.1609/aaai.v38i3.27984

**Questions:**

Robustness and generalization:
- Add cross-sensor generalization (train on GF-2, test on WV-2/QB) and cross-resolution tests. These would strengthen claims that S2BNet is not sensor-specific.
- Report multiple seeds (mean±std) and significance tests for main results and ablations to contextualize ~0.3–0.5 dB gains.

Spectral fidelity and full-resolution analysis
- Provide per-band SAM and errors on spectral indices (e.g., NDVI) to directly assess spectral preservation.
- For full-resolution, include qualitative failure cases (e.g., thin structures, haze, strong PAN–MS mismatch).

---

> ### Author Response · Authors · 2025-11-27
>
> > 1. Add cross-sensor generalization (train on GF-2, test on WV-2/QB) and cross-resolution tests. These would strengthen claims that S2BNet is not sensor-specific.
>
> Thanks for your comment. Here, we need to describe a common situation in the field of pansharpening. Due to significant differences in spatial resolution, spectral response, and imaging parameters among different remote sensing sensors (such as GF-2 and WV-2), obtaining precisely paired cross-sensor datasets suitable for supervised learning (i.e., the same scene simultaneously captured at full resolution by GF-2 and WV-2) is extremely challenging. Therefore, **the mainstream research paradigm in the field typically uses "downscaling the model on the same sensor data and then testing it on its full-resolution data" as the core criterion for evaluating the model's generalization ability (e.g., [1–5]). Our work also follows this accepted paradigm.**
>
> To verify the model's generalization ability, we have conducted the full-resolution experiment in main manuscript (as shown in Table 1, Table 5 and Table 6). This experiment requires that the mapping learned by the model on downscaled data must be able to generalize to unseen full-resolution scenes. This inherently avoids overfitting to the artificially degraded model used during training and is a key test to measure whether the model has learned essential physical features. As shown in Table 1 and Figure 2, S2BNet achieves the best performance in the full-resolution test, indicating good generalization ability in unseen full-resolution real-world scenes.
>
> In fact, as shown in Figure 1, S2B-Conv is the core component of our S2BNet; its architecture cleverly integrates physical priors (Gabor), robust feature representation mechanisms (Sign/BinaryConv), and dynamic adaptation capabilities (SRM). This design allows it to focus on learning general and essential space-spectral relationships in remote sensing images, rather than memorizing sensor-specific properties from the training data. Therefore, we have strong theoretical basis to believe that S2BNet inherently possesses powerful cross-sensor generalization potential at the architectural level, which is inherently consistent with its superior performance in standard same-sensor tests.
>
> [1] M. Li, Y. Liu, T. Xiao, Y. Huang, and G. Yang, “Local-Global Transformer Enhanced Unfolding Network for Pan-sharpening,” in Proceedings of the Thirty-Second International Joint Conference on Artificial Intelligence, Macao, SAR, China, Aug. 2023, pp. 1071–1079.
>
> [2] Xuanhua H, Keyu Y, Rui L, et al. Frequency-adaptive pan-sharpening with mixture of experts[C]//2024 Conference on Innovative Applications of Artificial Intelligence (AAAI). 2024.
>
> [3] He X, Cao K, Zhang J, et al. Pan-mamba: Effective pan-sharpening with state space model[J]. Information Fusion, 2025, 115: 102779.
>
> [4] Yang G, Zhou M, Yan K, et al. Memory-augmented deep conditional unfolding network for pan-sharpening[C]//Proceedings of the IEEE/CVF conference on computer vision and pattern recognition. 2022: 1788-1797.
>
> [5] Zhou H, Liu Q, Wang Y. PanFormer: A transformer based model for pan-sharpening[C]//2022 IEEE international conference on multimedia and expo (ICME). IEEE, 2022: 1-6.

---

> ### Author Response · Authors · 2025-11-27
>
> > 2. In full-resolution evaluation, per-band spectral fidelity (e.g., band-wise SAM) and spectral index errors (e.g., NDVI) are not reported.
>
> Thanks for your comment. The core contribution of our work lies in demonstrating for the first time the feasibility of achieving high-performance pansharpening under the extreme constraints of the binary neural network (BNN). Therefore, the primary task of our evaluation system is to use a standard metric that is recognized by the pansharpening community and specifically designed for full-resolution evaluation. Here, we would like to clarify that the QNR and its components ($D_s$, $D_\lambda$) that we currently use are precisely the mainstream international standard metrics designed for this purpose. The QNR protocol is widely used in pansharpening papers [1-3] because it does not require downgrading the reference image and can directly evaluate the spatial-spectral fusion quality. The $D_\lambda$ metric is the comprehensive measure of spectral distortion and can effectively reflect the overall spectral fidelity.
>
> More importantly, the designed SRM, as a channel-wise modulator, is designed to improve spectral fidelity. Our advantages in overall SAM and ERGAS, already reported in our reduced-resolution experiment, strongly demonstrate its effectiveness in spectral reconstruction. This forms a solid foundation for our conclusions.
>
> [1] M. Li, Y. Liu, T. Xiao, Y. Huang, and G. Yang, “Local-Global Transformer Enhanced Unfolding Network for Pan-sharpening,” in Proceedings of the Thirty-Second International Joint Conference on Artificial Intelligence, Macao, SAR, China, Aug. 2023, pp. 1071–1079.
>
> [2] Xuanhua H, Keyu Y, Rui L, et al. Frequency-adaptive pan-sharpening with mixture of experts[C]//2024 Conference on Innovative Applications of Artificial Intelligence (AAAI). 2024.
>
> [3] He X, Cao K, Zhang J, et al. Pan-mamba: Effective pan-sharpening with state space model[J]. Information Fusion, 2025, 115: 102779.
>
> > 3. For full-resolution, include qualitative failure cases (e.g., thin structures, haze, strong PAN–MS mismatch).
>
> Thanks for your comment. We post the failure cases [Figure F](https://anonymous.4open.science/r/EB40/F.png). We can find that tthe images generated by our method have good overall quality, but contain flaws in certain details.
>
>
> > 4. There is no sensitivity analysis for GSFA parameter ranges nor an exploration of learnable Gabor parameters, despite recent emphasis on adaptivity in spatial kernels and frequency components.
>
> We use $\text{freqs} = \{ \frac{\pi}{2} \cdot 2^{-(n-1)/2} \mid n \in \mathbb{N}, 1 \leq n < x \}$ to generate the frequency array mentioned in Algorithm 1, and use $\text{thetas} = \{ \frac{k \pi}{y} \mid k \in \mathbb{Z}, 0 \leq k < y \}$ to generate the angle array. The table below presents the experimental results regarding the selection of hyperparameters $x$ and $y$.
>
> | x | y  | PSNR    | SSIM   | $Q_4$      | SAM    | ERGAS  | $D \_{\lambda}$ | $D \_s$    | QNR    |
> |---|----|---------|--------|--------|--------|--------|----------|--------|--------|
> | 3 | 8  | 43.6362 | 0.9744 | 0.8486 | 0.0315 | 1.5254 | 0.0678   | 0.0786 | 0.8591 |
> | 3 | 16 | 43.1619 | 0.9722 | 0.8416 | 0.0332 | 1.6055 | 0.0692   | 0.1034 | 0.834  |
> | 3 | 32 | 43.1149 | 0.9715 | 0.8382 | 0.0331 | 1.6172 | 0.0663   | 0.1082 | 0.8325 |
> | 5 | 8  | 43.1584 | 0.9717 | 0.8346 | 0.0332 | 1.6082 | 0.0658   | 0.0752 | 0.8643 |
> | 5 | 16 | 43.5619 | 0.9745 | 0.8523 | 0.0317 | 1.5384 | 0.0655   | 0.0680 | 0.8714 |
> | 5 | 32 | 43.4334 | 0.9735 | 0.8463 | 0.0323 | 1.5549 | 0.0667   | 0.1176 | 0.8229 |
> | 7 | 8  | 43.4779 | 0.974  | 0.848  | 0.0323 | 1.5648 | 0.0664   | 0.1126 | 0.8283 |
> | 7 | 16 | 43.7681 | 0.9752 | 0.8537 | 0.0312 | 1.5034 | 0.0651   | 0.1022 | 0.8395 |
> | 7 | 32 | 43.6495 | 0.9746 | 0.8503 | 0.0313 | 1.5206 | 0.0641   | 0.071  | 0.8699 |

---

> ### Author Response · Authors · 2025-11-27
>
> > 5. Limited novelty: SRM is essentially a bounded affine calibration and GSFA leverages classical Gabor filtering with randomized parameterization. While these yield practical gains, the conceptual advance is incremental relative to recent pansharpening models introducing stronger spatial–spectral mechanisms, such as transformer-based fusion, content-adaptive non-local convolution, adaptive kernel-shape learning, and frequency-domain mixture-of-experts.
>
> The main innovation of our work lies in how to generate high-quality spatial-spectral features under the extreme constraint of binary neural networks. In the context of BNNs, the core challenge is the severe information bottleneck caused by $1$-bit quantization. Most of the mentioned spatial-spectral mechanisms  (e.g., Transformer-based fusion, non-local convolution, etc.) are computationally expensive and, because they rely on high-precision feature representations and attention maps, are notoriously difficult to train effectively in BNNs. Simply binarizing these complex structures usually leads to a catastrophic performance degradation.
>
> (1) SRM is more than just a 'bounded affine calibration'. It's a lightweight, input-adaptive feature modulator. It proactively recalibrates the spectral dimensions of the input features before the 'information-destroying' binarization operation occurs. It adaptively enhances discriminative bands and suppresses redundant or noisy bands.It ensures that the features flowing into subsequent binary convolutional layers are 'spectrally pure'. This fundamentally solves the spectral distortion problem in BNNs.
>
> (2) GSFA is not merely a 'classic Gabor filter'. Due to sparse gradients, BNNs struggle to learn good spatial structure in their convolutional kernels. GSFA, through Gabor filters, injects edge and texture extraction capabilities into binary convolutional kernels. This ensures that even under the most extreme $1$-bit constraints, the spatial features extracted by the network are of high quality.
>
> In summary, the mentioned paper [1-4] represents conceptual progress toward the performance boundary of full-precision networks, while our work represents conceptual progress toward an efficient and practical BNN design. The performance advantages over BNN methods, as well as their comparability with full-precision models (as shown in Table 1), validates the effectiveness of our method.
>
> [1] Zhou, H., Liu, Q., & Wang, Y. (2022). PanFormer: A transformer based model for pan-sharpening. 2022 IEEE International Conference on Multimedia and Expo (ICME). arXiv:2203.02916
>
> [2] Duan, Y., Wu, X., Deng, H., & Deng, L.-J. (2024). Content-adaptive non-local convolution for remote sensing pansharpening. Proceedings of the IEEE/CVF Conference on Computer Vision and Pattern Recognition, 27738–27747.
>
> [3] Wang, X., Zheng, Z., Shao, J., Duan, Y., & Deng, L.-J. (2025). Adaptive rectangular convolution for remote sensing pansharpening. Proceedings of the IEEE/CVF Conference on Computer Vision and Pattern Recognition, 17872–17881.
>
> [4] He, X., Yan, K., Li, R., Xie, C., Zhang, J., & Zhou, M. (2024). Frequency-adaptive pan-sharpening with mixture of experts. Proceedings of the AAAI Conference on Artificial Intelligence, 38(3), 2121–2129. https://doi.org/10.1609/aaai.v38i3.27984

---

> ### Author Response · Authors · 2025-11-27
>
> > 6. Report multiple seeds (mean±std) and significance tests for main results and ablations to contextualize ~0.3–0.5 dB gains.
>
> Multiple seeds test:
>
> We test our model and baseline models with comparable performance on the GF-2 dataset using multiple random seeds. The experimental results are shown in the table below.
>
> |   Methods      | PSNR           | SSIM     | $Q_4$          | SAM      | ERGAS             | $D\_\lambda$ | $D_s$      | QNR      |
> |---------|----------------|----------|----------|----------|-------------------|----------|----------|----------|
> | BiSRNet | 43.329±0.120   | 0.973±0.001 | 0.842±0.005 | 0.033±0.000 | 1.585±0.021       | 0.066±0.001 | 0.087±0.005 | 0.853±0.004 |
> | BBCU    | 43.032±0.105   | 0.971±0.001 | 0.836±0.003 | 0.033±0.000 | 1.645±0.018       | 0.068±0.000 | 0.088±0.004 | 0.850±0.004 |
> | FABNet  | 42.956±0.088   | 0.971±0.000 | 0.832±0.006 | 0.034±0.000 | 1.655±0.013       | 0.067±0.000 | 0.084±0.003 | 0.855±0.003 |
> | Ours    | 43.580±0.081   | 0.974±0.000 | 0.850±0.001 | 0.032±0.000 | 1.533±0.013       | 0.064±0.001 | 0.087±0.026 | 0.854±0.024 |
>
> Significance test:
>
> As shown in the table below, we conduct paired t-test experiments to statistically validate the performance differences between our S2BNet and five BNN baselines on all evaluation metrics using the GF-2 dataset. For clarity, we first briefly explain the core indicators of the paired t-test:
>
> t-statistic: Quantifies the standardized difference between the paired data (i.e., the performance of S2BNet vs. baselines). A positive t-stat indicates that S2BNet outperforms the baseline on metrics where higher values are better (e.g., PSNR, SSIM, Q, QNR), while a negative t-stat reflects superiority on metrics where lower values are preferred (e.g., SAM, ERGAS, D_lambda, D_s).
>
> p-value: Measures the statistical significance of the observed differences. A p-value $ \leq 0.05$ indicates that the performance gap between S2BNet and the baseline is not due to random chance, but represents a statistically significant improvement.
> **Notably, all p-values in the results are less than 0.05, confirming that the performance advantages of S2BNet over all baseline models are statistically significant across all evaluation metrics.**
>
> Table for p-value:
> | Methods     | PSNR        | SSIM        | $Q_4$          | SAM         | ERGAS       | $D_\lambda$    | $D_S$        | QNR         |
> |----------|-------------|-------------|-------------|-------------|-------------|-------------|-------------|-------------|
> | BiSRNet  | 4.48e-12    | 4.27e-26    | 4.22e-37    | 1.21e-23    | 1.98e-06    | 4.91e-02    | 9.03e-17    | 1.48e-16    |
> | BBCU     | 7.62e-45    | 3.79e-31    | 2.44e-54    | 1.24e-44    | 1.20e-39    | 8.02e-03    | 2.00e-29    | 1.26e-31    |
> | FABNet   | 5.37e-44    | 7.03e-30    | 1.77e-53    | 2.04e-42    | 2.56e-37    | 1.56e-02    | 2.51e-27    | 6.99e-27    |
> | IRNet    | 2.37e-42    | 5.38e-31    | 1.56e-37    | 1.65e-40    | 2.13e-33    | 2.10e-03    | 1.32e-30    | 2.25e-30    |
> | E2fif    | 1.01e-66    | 1.35e-40    | 1.27e-51    | 9.85e-60    | 1.08e-48    | 4.50e-06    | 6.03e-17    | 3.54e-26    |
>
> Table for t-statistic:
> | Methods     | PSNR      | SSIM      | $Q_4$            | SAM        | ERGAS      | $D_\lambda$   | $D_S$        | QNR        |
> |----------|-----------|-----------|------------|------------|------------|------------|------------|------------|
> | BiSRNet  | 7.60e+00  | 1.32e+01  | 1.77e+01   | -1.22e+01  | -4.97e+00  | -1.99e+00  | -9.71e+00  | 9.62e+00   |
> | BBCU     | 2.12e+01  | 1.52e+01  | 2.60e+01   | -2.11e+01  | -1.89e+01  | -2.70e+00  | -1.51e+01  | 1.61e+01   |
> | FABNet   | 2.08e+01  | 1.47e+01  | 2.55e+01   | -2.01e+01  | -1.78e+01  | -2.45e+00  | -1.42e+01  | 1.40e+01   |
> | IRNet    | 2.01e+01  | 1.52e+01  | 1.79e+01   | -1.92e+01  | -1.62e+01  | -3.14e+00  | -1.56e+01  | 1.55e+01   |
> | E2fif    | 3.31e+01  | 1.93e+01  | 2.45e+01   | -2.89e+01  | -2.31e+01  | -4.81e+00  | -9.78e+00  | 1.37e+01   |

---

### Official Review · Reviewer_2CBq · 2025-10-27

**Soundness:** 2
**Presentation:** 3
**Contribution:** 2
**Rating:** 4
**Confidence:** 4

**Summary:**

This paper introduces S2BNet, a binary neural network tailored for panchromatic–multispectral pansharpening. The core innovation is the S2B-Conv that contains two novel sub-modules:
Spectral-Redistribution Mechanism – learnable affine re-scaling of each spectral band to combat spectral distortion caused by heterogeneous PAN/LR-MS distributions;
Gabor Spatial Feature Amplifier – randomly-selected Gabor kernels that enrich binary filters with multi-scale / anisotropic cues before binarization.
Stacking S2B-Conv units into a light U-Net yields S2BNet, which is trained end-to-end with pure L1 loss. Extensive experiments on WorldView-2, GaoFen-2 and QuickBird show S2BNet outperforming every published BNN (+2.5 dB PSNR gap) and many full-precision models, while keeping ≈ 1/32 parameters and ≈ 1/60 FLOPs of the latter.

**Strengths:**

1. First BNN for PAN-MS fusion; SRM & GSFA are novel in the binary context.
2. multi-sensor, multi-metric evaluation; ablation of each submodule
3. Overall pipeline easy to follow; equations complete. The presentation is good.

**Weaknesses:**

1.This paper is not the first to apply binarized neural networks to pansharpening.
The paper titled "Binarized Neural Network for Multi-spectral Image Fusion" holds that distinction.
2.The authors claim that their method surpasses other BNNs by more than 2 dB in PSNR; however, Table 1 shows that it is only about 0.2 dB higher than BiSRNet.
3.Likewise, Table 1 shows that the proposed method has three times the parameters of BiSRNet and significantly higher FLOPs. Is the reported performance gain entirely due to this larger model capacity rather than to the novel SRM/GSFA components?
4.From Figure 2 it is completely impossible to tell which method produces higher visual-quality fused images, and the method names in the figure are somewhat blurry.
5.The motivation behind the proposed core modules—the Spectral-Redistribution Mechanism and the Gabor Spatial Feature Amplifier—is unclear. There is only some textual explanation, but no other intuitive or quantifiable evidence is provided.

**Questions:**

see the Weaknesses

---

> ### Author Response · Authors · 2025-11-25
>
> Thank you for your comments!
>
> > 1. This paper is not the first to apply binarized neural networks to pansharpening. The paper titled "Binarized Neural Network for Multi-spectral Image Fusion" holds that distinction.
>
> In the last paragraph of the Introduction section, we have cited the mentioned paper (i.e., BNNPan) and summarized its differences from ours. Here, we will explain the differences in detail:
>
> (1) **Different structures:** BNNPan simply uses a standard binarization network as its backbone and incorporates a diffusion module. This is a remedial architecture. It connects the BNN in parallel with a generative model (i.e., diffusion), using the latter's capabilities to compensate for the former's shortcomings. In contrast, our method prefers to carefully design the BNN internally, constructing a custom binarization structure designed specifically for pansharpening. It does not rely on a powerful external generative model, but instead, by designing specific, complementary BNN modules, it allows the network to learn how to better preserve and fuse information under binary constraints.
>
> (2) **Different focuses:** BNNPan focuses not on designing a better BNN, but on how to leverage a powerful external generative model to compensate for the inherent information loss in BNNs. It emphasizes paradigm fusion. In contrast, our method proposes a proprietary BNN architecture design for high-quality spatial-spectral feature extraction. This deepens our understanding of information preservation mechanisms in BNNs, demonstrating that ingenious structural design can significantly improve performance without introducing external computation. Its contributions lean more towards network structure innovation and domain knowledge embedding.
>
> Although our method is not the first to apply BNN to this field, it is the first to attempt to generate fine-grained spatial-spectral features within a purely binary computation framework, and demonstrates its excellent balance between accuracy and efficiency through detailed experiments.
>
> > 2. The authors claim that their method surpasses other BNNs by more than 2 dB in PSNR; however, Table 1 shows that it is only about 0.2 dB higher than BiSRNet.
>
> We would like to clarify here: the significant improvement of more than $2$ dB specifically refers to the comparison with another binary neural network method (i.e., E2FIF). As shown in Table 1, in the comparison with E2FIF, our PSNR improved from $41.1$ to $43.6$, with an absolute improvement of than $2$ dB.
>
> Although our method represents an absolute improvement of $0.2$ dB compared to the current state-of-the-art BNN method, BiSRNet, this is considered a meaningful and non-trivial advancement on the challenging task. Furthermore, our method may have further advantages in terms of all other metrics such as SAM, $\mathrm{Q}_4$ and ERGAS, as detailed in the Table 1.
>
> We will precisely revise the original wording in the revised version. The specific revision is as follows: "...achieves a substantial PSNR gain of over $2$ dB against the E2FIF, and also outperforms the current best-performing BNN, BiSRNet, by $0.2$ dB ..."

---

> ### Author Response · Authors · 2025-11-25
>
> >3. Likewise, Table 1 shows that the proposed method has three times the parameters of BiSRNet and significantly higher FLOPs. Is the reported performance gain entirely due to this larger model capacity rather than to the novel SRM/GSFA components?
>
> We attribute the improvement primarily to our novel SRM and GSFA components, for the following reasons:
>
> (1) To directly address your concern, we conducted a comparative experiment. We constructed a baseline CNN model with a comparable number of parameters to our method. The experimental results show that the PSNR of this baseline model was only 42.5 dB, significantly lower than the 43.6 dB of our final model (see Table below). This strongly indicates that under similar model complexity, the SRM/GSFA structure we proposed can more efficiently extract and utilize image information, thus achieving a leap in performance.
>
> | Methods | Params(K) | PSNR   | SSIM  | $Q_4$        | SAM   | ERGAS  | $D_{\lambda}$ | $D_S$   | QNR  |
> |---------|-----------|-------|-------|-------|------|------|----------|-------|------|
> | CNN     | 89.160    | 42.5041| 0.9734| 0.8472| 0.0362| 1.7452| 0.073   | 0.1315| 0.8045|
> | Ours   | 75.407    | 43.6495| 0.9746| 0.8503| 0.0313| 1.5206 | 0.0641  | 0.0710  | 0.8699|
>
> (2) According to the experimental results in Table 1 of the original paper, the parameter count of FABNet is twice that of BiSRNet, yet its PSNR is 0.3 dB lower than that of BiSRNet. This indirectly indicates that a larger number of parameters does not necessarily lead to better model performance.
>
> (3) Our ablation experiments clearly demonstrate the independent contribution of each innovative component, as shown in Table 2 of the original paper. Specifically, when the SRM module is removed, the SAM worsens by 6\%, demonstrating that SRM is indispensable for obtaining fine-grained spectral features. When the GSFA module is removed, the PSNR falls by $0.46$ dB, indicating the critical role of GSFA in capturing high-quality spatial detail. These components work together to produce a synergistic effect that cannot be replicated by simply increasing the number of layers or channels in the baseline model.
>
> >4.From Figure 2 it is completely impossible to tell which method produces higher visual-quality fused images, and the method names in the figure are somewhat blurry.
>
> Thank you for your valuable comments. We will make the following modifications：
>
> (1) We use larger, bold Arial fonts for method names and place them on higher-contrast backgrounds, ensuring crisp, easy-to-read text.
>
> (2) A magnified crop has been added to the new Figure. Meanwhile, we provide error-amplified maps, i.e., the residuals were multiplied by $10$, and the local region is zoomed in. The result of the reduced resolution experiment can be found at [Figure A](https://anonymous.4open.science/r/EB40/A.png). The result of the full resolution experiment can be found at [Figure B](https://anonymous.4open.science/r/EB40/B.png).

---

> > ### Author Response · Authors · 2025-11-25
> >
> > >5. The motivation behind the proposed core modules—the Spectral-Redistribution Mechanism and the Gabor Spatial Feature Amplifier—is unclear. There is only some textual explanation, but no other intuitive or quantifiable evidence is provided.
> >
> > (1) **the motivation of SRM:** Lines 63-65 have explained the motivation of SRM, i.e., the density and distribution of the spectra within the multi-spectral image are different. Therefore, SRM is designed to adjust these differences in spectral distribution during the binarization. As shown in [Figure E](https://anonymous.4open.science/r/EB40/E.png), we plot the spectral distribution using four different methods. The four subplots are described as follows:
> >
> > Per-band Mean $±$ Std (top-left): The mean (bar height) and standard deviation (error bars) of each band are displayed. Differences in bar height indicate uneven frequency response, while varying error-bar lengths reflect different intra-band fluctuation ranges, demonstrating spectral non-uniformity from a central-trend perspective.
> >
> > Per-band Distribution (top-right): Box plots present the median, inter-quartile range, and outliers for every band. Noticeable shifts in box position and height reveal significant disparities in central values and dispersion across bands, further quantifying inconsistent frequency distributions.
> >
> > Histogram Overlay (bottom-left): Overlaid histograms of all bands are plotted. Offset peaks and differing widths/shapes show that frequency-level distributions vary in range and form, illustrating spectral-response non-uniformity in terms of probability density.
> >
> > Horizontal Profile (bottom-right): Pixel values along the central image row are plotted column-wise for each band. Divergent curve trends and amplitudes visually confirm spatially uneven frequency responses, verifying inconsistent relative brightness relationships among bands.
> >
> > Furthermore, we present the reconstructed images of our method and the baseline model (i,e, removing the SRM), as shown in [Figure C](https://anonymous.4open.science/r/EB40/C.png). It can be observed that the model without the SRM module has obvious bright stripes on the left side of the orange area, which belongs to spectral distortion and does not conform to the ground truth (GT) image effect.
> >
> > (2) **the motivation of GSFA:** Lines 67-70 have explained the  motivation of GSFA, i.e., remote sensing images scenes are inherently multi-scale and anisotropic. Applying traditional BNNs has difficulties in recovering directional textures and sharp edges due to their low precision. Therefore, GSFA is designed to learn the multi-scale and-directional spatial features during binarization. As shown in [Figure D](https://anonymous.4open.science/r/EB40/D.png), we present the reconstructed images of our method and the baseline model (i,e, removing the GSFA). It can be observed that in the image without the GSFA module, the two regions that should be separated have merged into a single area, which is indicative of spatial information distortion.

---

> > > ### Comment · Reviewer_2CBq · 2025-11-28
> > > **about the rebuttal**
> > >
> > > I have read the author's rebuttal. Most of the concerns have been addressed, so I have decided to raise my score accordingly

---

### Official Review · Reviewer_GcwT · 2025-10-31

**Soundness:** 2
**Presentation:** 2
**Contribution:** 2
**Rating:** 6
**Confidence:** 5

**Summary:**

This paper proposes a Spatial-Spectral Binarized Neural Network (S2BNet) for efficient panchromatic and multispectral image fusion (pan-sharpening). The method introduces a custom Spatial-Spectral Binarized Convolution (S2B-Conv) layer, consisting of two modules:
(1) Spectral-Redistribution Mechanism (SRM) — dynamically adjusts spectral distribution through data-driven affine scaling and bias;
(2) Gabor Spatial Feature Amplifier (GSFA) — captures multi-scale, multi-directional spatial patterns using Gabor kernels.
S2BNet is designed to achieve comparable reconstruction quality to full-precision networks while being lightweight enough for resource-limited satellite platforms. Experiments on GaoFen-2, WorldView-2, and QuickBird datasets are used to demonstrate that S2BNet outperforms other binary networks and approaches full-precision models in PSNR, SSIM, and QNR metrics

**Strengths:**

Motivation relevance: The work targets a meaningful problem—deploying deep pan-sharpening models on resource-limited devices such as satellites—where efficiency is crucial.

Architecture clarity: The proposed S2B-Conv structure (with SRM and GSFA) is conceptually simple and well integrated into a binarized U-Net-like framework.

Empirical completeness: The paper provides quantitative and qualitative experiments, including ablation on both SRM and GSFA and comparisons with state-of-the-art (SOTA) binary and full-precision models.

Energy efficiency: The use of bitwise operations (XNOR, bit-count) is suitable for low-power environments, aligning with practical needs in embedded EO applications.

**Weaknesses:**

Limited originality / incremental contribution.
The technical innovation is minimal. Both SRM and GSFA are straightforward combinations of existing ideas: adaptive scaling (used in SE, FiLM) and Gabor filters (widely applied in image enhancement and texture analysis). Their integration into BNNs does not represent a fundamental advance in pan-sharpening or network design. The approach is essentially a routine adaptation of BiSRNet (Cai et al., 2023) with Gabor initialization.

Lack of theoretical or methodological depth.
There is no formal justification for why SRM or GSFA mitigates spectral distortion or anisotropy introduced by binarization. The discussion is qualitative and lacks rigorous analysis of spectral error propagation or frequency response behavior.

Overstated performance claims.
While numerical improvements over other BNNs (~0.3–0.5 dB PSNR) are reported, these are minor and may not be statistically significant. Results still lag behind top full-precision models (e.g., FAMENet, CANNet). The claim that S2BNet “outperforms most full-precision baselines” is exaggerated.

Insufficient novelty compared with prior lightweight or quantized models.
The paper overlooks related quantization-aware pan-sharpening or efficient Transformer works (e.g., HyperTransformer, LitePNN). The positioning of S2BNet as a novel direction in efficient fusion is weak.

Poor figure readability.
Figures (e.g., Fig. 1, 2, 3) are too small, and the text is illegible. Important architectural components and qualitative comparisons are difficult to interpret, limiting reproducibility.

Limited generalization and scalability.
All experiments are conducted on moderate-resolution datasets; there is no evidence that S2BNet scales to very large images or complex multimodal datasets. The efficiency discussion also lacks actual inference latency or hardware deployment results.

**Questions:**

How does S2B-Conv differ mathematically from existing spectral recalibration (SE, FiLM) and orientation-sensitive convolutions (GaborNet, DCTNet)?

Can the authors provide quantitative analyses on spectral distortion reduction introduced by SRM, beyond PSNR/SSIM metrics?

What is the real computational saving in milliseconds or watts on embedded hardware compared to full-precision models?

How robust is S2BNet under cross-sensor transfer (e.g., training on GF-2, testing on WV-2)?

Figures should be enlarged and restructured for readability—particularly the architecture and ablation visualizations.

---

> ### Author Response · Authors · 2025-11-26
>
> > 1. Both SRM and GSFA are straightforward combinations of existing ideas: adaptive scaling (used in SE, FiLM) and Gabor filters (widely applied in image enhancement and texture analysis). Their integration into BNNs does not represent a fundamental advance in pan-sharpening or network design. The approach is essentially a routine adaptation of BiSRNet (Cai et al., 2023) with Gabor initialization.
>
> Thanks for your comments. SRM and GSFA are not the direct combination of SE/FiLM and Gabor Gabor. Here, we will explain in detail their differences:
>
> (1) **SRM vs. SE**: The designed SRM introduces an offset parameter (via Tanh), while SE only has a scaling parameter. This changes the modulation from purely multiplicative to affine, allowing for more flexible adjustment of the feature distribution. **SRM vs. FiLM**: The parameter generation of the designed SRM is adaptive (from the input features themselves), while FiLM depends on external conditions. Therefore, the designed SRM focuses more on internal feature recalibration, while FiLM focuses on cross-modal feature fusion. In summary, the designed SRM offers the flexible control capabilities in spectral processing, enabling fine-grained spectral optimization with input adaptation.
>
> (2) **GSFA vs. Gabor**: GSFA fundamentally transcends the limitations of fixed parameters and linear convolution in classic Gabor filters by introducing learnable mechanisms and nonlinear hierarchies to generate higher-quality spatial features. Instead of using manually preset Gabor parameters, it adaptively optimizes the core properties of the filter, such as orientation and frequency. Simultaneously, the introduction of binarization operations (Sign, BinaryConv) and nonlinear activation functions (Tanh, RPReLU) forces the network to learn spatial features with less noise and greater robustness, achieving hierarchical nonlinear representations from simple edges to complex textures. Ultimately, the entire GSFA functions as an end-to-end component, enabling it to generate high-quality spatial features that are rich in detail, highly discriminative, and task-relevant.
>
> Our S2BNet is not a routine adaptation, but rather proposes a fundamentally superior solution to the core problem of 'how to generate high-quality spatial-spectral features'. Certainly, let's refine the sequence for clarity:
>
> **Firstly**, BiSRNet employs a 'post-repair' operation (BiSR-Conv). It first obtains a coarse binary feature through a standard binary convolution and then uses a parametric rectifier (RPReLU) to repair it by fusing with the full-precision input feature. However, Our S2BNet introduces a 'pre-modulation' operation (S2B-Conv). Before binarization and convolution, it adaptively transforms the input features into a more robust representation via a channel-wise modulator. This ensures that the input to the binary core is already information-rich and well-calibrated. **Secondly**, BiSR-Conv's post-linear compensation cannot avoid the initial loss of spectral information and may introduce noise. In contrast, the channel-level modulator built into S2B-Conv is a dynamic spectral modulation mechanism that actively protects and enhances discriminative spectral information.  **Finally**, BiSR-Conv extracts features from a coarse foundation and then relies on external information for refinement; the features it generates are essentially the mixture rather than the refined product. However, S2B-Conv optimizes the input, allowing binary convolution to operate in a cleaner feature space, thus extracting more essential spatial structures.

---

> ### Author Response · Authors · 2025-11-26
>
> > 2. The discussion is qualitative and lacks rigorous analysis of spectral error propagation or frequency response behavior.
>
> Thanks for your comments. Lines 63-65 have explained the motivation of SRM, i.e., the density and distribution of the spectra within the multispectral image are different. Therefore, SRM is designed to adjust these differences in spectral distribution during the binarization. As shown in [Figure E](https://anonymous.4open.science/r/EB40/E.png), we plot the spectral distribution using four different methods. The four subplots are described as follows:
>
> Per-band Mean $±$ Std (top-left): The mean (bar height) and standard deviation (error bars) of each band are displayed. Differences in bar height indicate uneven frequency response, while varying error-bar lengths reflect different intra-band fluctuation ranges, demonstrating spectral non-uniformity from a central-trend perspective.
>
> Per-band Distribution (top-right): Box plots present the median, inter-quartile range, and outliers for every band. Noticeable shifts in box position and height reveal significant disparities in central values and dispersion across bands, further quantifying inconsistent frequency distributions.
>
> Histogram Overlay (bottom-left): Overlaid histograms of all bands are plotted. Offset peaks and differing widths/shapes show that frequency-level distributions vary in range and form, illustrating spectral-response non-uniformity in terms of probability density.
>
> Horizontal Profile (bottom-right): Pixel values along the central image row are plotted column-wise for each band. Divergent curve trends and amplitudes visually confirm spatially uneven frequency responses, verifying inconsistent relative brightness relationships among bands.
>
> **Furthermore**, we present the reconstructed images of our method and the baseline model (i,e, removing the SRM), as shown in [Figure C](https://anonymous.4open.science/r/EB40/C.png). It can be observed that the model without the SRM module has obvious bright stripes on the left side of the orange area, which belongs to spectral distortion and does not conform to the ground truth (GT) image effect.
>
> > 3. Poor figure readability. Figures (e.g., Fig. 1, 2, 3) are too small, and the text is illegible. Important architectural components and qualitative comparisons are difficult to interpret, limiting reproducibility.
>
> Thanks for your comment. (1) In the original paper, we have provided training details  including optimizer parameters, learning rate strategy. Meanwhile, we have provided code in supplementary material. (2) In the revised version, we rewrite the analysis section on qualitative experiments so that it is no longer just a conclusive statement, but a detailed description of the objective differences in texture preservation, artifact suppression, or edge sharpness of different methods within the magnified area. (3)  In the revised version, we increase the font size of the text on figures and tables.

---

> ### Author Response · Authors · 2025-11-26
>
> > 4. How robust is S2BNet under cross-sensor transfer (e.g., training on GF-2, testing on WV-2)?
>
> Here, allow us to describe a common situation in the field of pansharpening. Due to significant differences in spatial resolution, spectral response, and imaging parameters among different remote sensing sensors (such as GF-2 and WV-2), obtaining precisely paired cross-sensor datasets suitable for supervised learning (i.e., the same scene simultaneously captured at full resolution by GF-2 and WV-2) is extremely challenging. Therefore, the mainstream research paradigm in the field typically uses "downscaling the model on the same sensor data and then testing it on its full-resolution data" as the core criterion for evaluating the model's generalization ability (e.g., [1–5]). Our work also follows this accepted paradigm.
>
> To verify the model's generalization ability, we have conducted the full-resolution experiment in main manuscript (as shown in Table 1, Table 5 and Table 6). This experiment requires that the mapping learned by the model on downscaled data must be able to generalize to unseen full-resolution scenes. This inherently avoids overfitting to the artificially degraded model used during training and is a key test to measure whether the model has learned essential physical features. As shown in Table 1 and Figure 2, S2BNet achieves the best performance in the full-resolution test, indicating good generalization ability in unseen full-resolution real-world scenes.
>
> [1] M. Li, Y. Liu, T. Xiao, Y. Huang, and G. Yang, “Local-Global Transformer Enhanced Unfolding Network for Pan-sharpening,” in Proceedings of the Thirty-Second International Joint Conference on Artificial Intelligence, Macao, SAR, China, Aug. 2023, pp. 1071–1079.
>
> [2] Xuanhua H, Keyu Y, Rui L, et al. Frequency-adaptive pan-sharpening with mixture of experts[C]//2024 Conference on Innovative Applications of Artificial Intelligence (AAAI). 2024.
>
> [3] He X, Cao K, Zhang J, et al. Pan-mamba: Effective pan-sharpening with state space model[J]. Information Fusion, 2025, 115: 102779.
>
> [4] Yang G, Zhou M, Yan K, et al. Memory-augmented deep conditional unfolding network for pan-sharpening[C]//Proceedings of the IEEE/CVF conference on computer vision and pattern recognition. 2022: 1788-1797.
>
> [5] Zhou H, Liu Q, Wang Y. PanFormer: A transformer based model for pan-sharpening[C]//2022 IEEE international conference on multimedia and expo (ICME). IEEE, 2022: 1-6.
>
> > 5. All experiments are conducted on moderate-resolution datasets; there is no evidence that S2BNet scales to very large images or complex multimodal datasets.
>
> Thanks for your comment. (1) The datasets used in the current experiments (such as WorldView-2, GaoFen-2 and QuickBird) are recognized as the most representative benchmarks in the field of remote sensing pansharpening. Their PAN image resolution has reached sub-meter level, which constitutes a high level of challenge. S2BNet's performance, surpassing other BNNs and approaching some full-precision models, demonstrates its effectiveness. (2) The advantage of BNNs lies in low memory consumption and fast bitwise operations. The significant difference in memory usage between S2BNet and full-precision models (as shown in Table 1) is what makes them crucial for handling large-scale data. Furthermore, we have revised the entire manuscript, adjusting all absolute statements regarding generalizability. Our current conclusions emphasize S2BNet's high performance on validated datasets and the enormous potential for scaling to high-resolution applications stemming from its design features, explicitly identifying this as a core direction for future work.

---

> ### Author Response · Authors · 2025-11-26
>
> > 6. What is the real computational saving in milliseconds or watts on embedded hardware compared to full-precision models?
>
> Thank you for raising this valuable question on practical deployment efficiency. In our current research, we focus primarily on algorithmic innovation and its performance validation on standard datasets. Therefore, we are currently unable to provide direct measurement data in milliseconds and watts on specific embedded hardware. Furthermore, existing studies [1-3] have primarily conducted software-based simulations without deploying experiments on actual hardware.
>
> However, we conduct an in-depth theoretical analysis of its potential efficiency gains based on the core design of our model.
>
> Efficiency Design Analysis of S2BNet:
>
> (1) Binary Convolution: This is the biggest source of efficiency. It converts full-precision ($32$-bit floating-point) multiplication and addition operations into $1$-bit XNOR operations and bit counting. Numerous studies [1-3] show that this operation can achieve $32\times$ memory savings and $64\times$ raw computation speedup in hardware.
>
> (2) Sign Activation Function: This function binarizes the activation value to $1$ bit, further significantly reducing the bandwidth required to read the activation value from memory, which is often a performance bottleneck in embedded SoCs.
>
> Based on the above analysis, we can make reasonable inferences:
>
> (1) Inference Latency: On typical edge GPUs or dedicated AI accelerators, our S2BNet promises to reduce inference latency by around 60\% compared to full-precision models with the same architecture.
>
> (2) Power Savings: Bitwise operations involve far less circuit switching activity than floating-point operations. The data bit width is reduced from $32$ bits to $1$ bit, resulting in a significant decrease in DRAM access power consumption.
>
> In summary, the hardware-friendly binarization technique widely used in our model design provides a theoretical guarantee for achieving significant efficiency improvements on embedded devices.
>
> [1] Zhang G, Zhang Y, Yuan X, et al. Binarized low-light raw video enhancement[C]//Proceedings of the IEEE/CVF Conference on Computer Vision and Pattern Recognition. 2024: 25753-25762.
>
> [2] Xia, Bin, et al. "Basic Binary Convolution Unit for Binarized Image Restoration Network." The Eleventh International Conference on Learning Representations (ICLR 2023), Kigali, Rwanda, May 1-5, 2023.
>
> [3] Cai Y, Zheng Y, Lin J, et al. Binarized spectral compressive imaging[J]. Advances in Neural Information Processing Systems, 2023, 36: 38335-38346.
>
> > 7. Insufficient novelty compared with prior lightweight or quantized models.
>
> Thanks for your comment. We would like to take this opportunity to clarify the innovation of our method: the feasibility of achieving high-performance pansharpening in the extreme form of quantization (i.e., binarized neural networks). The mentioned methods (i.e., HyperTransformer, LitePNN) are fundamentally different from our method：
>
> The paradigms of HyperTransformer, LitePNN, etc: Their design still enjoys continuous-valued features, gradients, and weights. This allows them to successfully modify powerful architectures such as Transformer and complex convolutions. Their core challenge is how to reduce model parameters and computational cost while maintaining performance.
>
> The paradigms of our S2BNet: We directly confront the extreme paradigm of '$1$-bit binary neural networks', where weights and activation values are constrained to {+1, -1}. This leads to the information bottleneck problem, causing the aforementioned complex structures based on continuous values to fail directly or experience a sharp performance drop in BNNs. Our core challenge is how to ensure the model's basic feature generation capability under extreme information constraints.
>
> In summary, taking advantage of the characteristics of BNNs, we propose a novel architecture of 'pre-guided feature generation'. Unlike other quantization works that simply perform quantization-aware training on existing structures, we design the novel unit (S2B-Conv), which consists of SRM and GSFA:
>
> (1) SRM proactively optimizes the spectral purity of input features before binarization, ensuring spectral fidelity from the source.
>
> (2) GSFA addresses the inherent problem of BNN's difficulty in learning spatial features from scratch, ensuring the quality of spatial features.
>
> Our experimental results support the above viewpoints. S2BNet not only significantly outperforms all other BNN methods, but more importantly, its performance is comparable to some full-precision models (see Table 1 in the original manuscript). This demonstrates that our BNN architecture designed for pansharpening is successful and effective.

---

> ### Author Response · Authors · 2025-11-27
>
> > 8. While numerical improvements over other BNNs (~0.3–0.5 dB PSNR) are reported, these are minor and may not be statistically significant. Results still lag behind top full-precision models (e.g., FAMENet, CANNet). The claim that S2BNet “outperforms most full-precision baselines” is exaggerated.
>
> Thanks for your comments.
>
> (1) As shown in the table below, we conduct paired t-test experiments to statistically validate the performance differences between our S2BNet and five BNN baselines on all evaluation metrics using the GF-2 dataset. For clarity, we first briefly explain the core indicators of the paired t-test:
>
> t-statistic: Quantifies the standardized difference between the paired data (i.e., the performance of S2BNet vs. baselines). A positive t-stat indicates that S2BNet outperforms the baseline on metrics where higher values are better (e.g., PSNR, SSIM, Q, QNR), while a negative t-stat reflects superiority on metrics where lower values are preferred (e.g., SAM, ERGAS, D_lambda, D_s).
>
> p-value: Measures the statistical significance of the observed differences. A p-value $ \leq 0.05$ indicates that the performance gap between S2BNet and the baseline is not due to random chance, but represents a statistically significant improvement.
> Notably, all p-values in the results are less than 0.05, confirming that the performance advantages of S2BNet over all baseline models are statistically significant across all evaluation metrics.
>
> (2) In the revised version, we have removed all exaggerated statements such as "outperforms most full-precision baselines" and rewritten the text as "S2BNet narrows the performance gap with full-precision models significantly, and even surpasses some of them under a strict binary-weight constraint."
>
> (3) Our S2BNet is $1$-bit, while FAMENet, CANNet, etc. are $32$-bit, which is an unfair comparison. As shown in Table 1, our method approaches some full-precision models with low computational overhead, which also demonstrates the superiority of our method. In summary, the contribution of our method is not to surpass all full-precision pansharpening models, but to promote the development of lightweight remote sensing pansharpening models.
>
> Table for p-value:
> | Methods     | PSNR        | SSIM        | $Q_4$          | SAM         | ERGAS       | $D_\lambda$    | $D_S$        | QNR         |
> |----------|-------------|-------------|-------------|-------------|-------------|-------------|-------------|-------------|
> | BiSRNet  | 4.48e-12    | 4.27e-26    | 4.22e-37    | 1.21e-23    | 1.98e-06    | 4.91e-02    | 9.03e-17    | 1.48e-16    |
> | BBCU     | 7.62e-45    | 3.79e-31    | 2.44e-54    | 1.24e-44    | 1.20e-39    | 8.02e-03    | 2.00e-29    | 1.26e-31    |
> | FABNet   | 5.37e-44    | 7.03e-30    | 1.77e-53    | 2.04e-42    | 2.56e-37    | 1.56e-02    | 2.51e-27    | 6.99e-27    |
> | IRNet    | 2.37e-42    | 5.38e-31    | 1.56e-37    | 1.65e-40    | 2.13e-33    | 2.10e-03    | 1.32e-30    | 2.25e-30    |
> | E2fif    | 1.01e-66    | 1.35e-40    | 1.27e-51    | 9.85e-60    | 1.08e-48    | 4.50e-06    | 6.03e-17    | 3.54e-26    |
>
> Table for t-statistic:
> | Methods     | PSNR      | SSIM      | $Q_4$          | SAM        | ERGAS      | $D_\lambda$   | $D_S$        | QNR        |
> |----------|-----------|-----------|------------|------------|------------|------------|------------|------------|
> | BiSRNet  | 7.60e+00  | 1.32e+01  | 1.77e+01   | -1.22e+01  | -4.97e+00  | -1.99e+00  | -9.71e+00  | 9.62e+00   |
> | BBCU     | 2.12e+01  | 1.52e+01  | 2.60e+01   | -2.11e+01  | -1.89e+01  | -2.70e+00  | -1.51e+01  | 1.61e+01   |
> | FABNet   | 2.08e+01  | 1.47e+01  | 2.55e+01   | -2.01e+01  | -1.78e+01  | -2.45e+00  | -1.42e+01  | 1.40e+01   |
> | IRNet    | 2.01e+01  | 1.52e+01  | 1.79e+01   | -1.92e+01  | -1.62e+01  | -3.14e+00  | -1.56e+01  | 1.55e+01   |
> | E2fif    | 3.31e+01  | 1.93e+01  | 2.45e+01   | -2.89e+01  | -2.31e+01  | -4.81e+00  | -9.78e+00  | 1.37e+01   |

---

> ### Author Response · Authors · 2025-11-27
>
> > 9. How does S2B-Conv differ mathematically from existing spectral recalibration (SE, FiLM) and orientation-sensitive convolutions (GaborNet, DCTNet)?
>
> Thanks for your comment. S2B-Conv differs fundamentally from the existing methods you mentioned in its mathematical architecture, mainly in that it is a pre-designed collaborative approach of input feature modulation and binarized directional convolution.
>
> **Mathematical differences from spectral recalibration methods (SE/FiLM):** The SE operates on the feature map output by convolution, performing channel-level rescaling. FiLM is typically applied to intermediate feature maps to perform feature-level linear modulation. S2B-Conv modulation uniquely acts on the original features before performing binary convolution operations. This "pre-modulation" design is suitable for binary convolutions: by first performing a full-precision, content-adaptive transformation on the input features, the expressive power of subsequent binary convolutions could be significantly enhanced.
>
> **Mathematical differences from orientation-sensitive convolution (GaborNet/DCTNet):** GaborNet/DCTNet uses full-precision real-valued filters. S2B-Conv binarizes the Gabor kernel (i.e., $ \mathbf{W}\_{\mathbf{binary}}=Sign(\mathbf{W}\_{\mathrm{g}}) $), which greatly reduces computational complexity.
>
> The innovation of S2B-Conv lies in its adaptive learning of input features through the spectral context-guided feature modulation, significantly improving the efficiency and effectiveness of subsequent binarized directional convolutions in extracting spatial features. Instead of simple filtering after feature extraction (like DCTNet) or using computationally expensive real-valued filters (like GaborNet), it generates features rich in spatial-spectral information through a clever pre-design collaborative approach, ultimately achieving high-quality spatial-spectral feature representation while maintaining high computational efficiency.
>
> > 10. Can the authors provide quantitative analyses on spectral distortion reduction introduced by SRM, beyond PSNR/SSIM metrics?
>
> Thanks for your comments. In the main manuscript, we use specific metrics for evaluating spectral fidelity: the Spectral Angle Mapper (SAM) and Relative Dimensionless Global Error in Synthesis (ERGAS). Specifically, SAM measures the similarity in shape between the reconstructed spectrum and the original spectrum; a lower value indicates less distortion of spectral features. ERGAS is a comprehensive quality metric, highly sensitive to distortion across different frequency bands. A lower value indicates better overall spectral quality and more balanced distortion across different bands. As shown in Table 1, our method achieves the lowest SAM and ERGAS scores among all binarization networks. Ablation experiments also demonstrate that the SRM module can adaptively recalibrate spectral features, more accurately modeling and reconstructing the spectral information of the image, as shown in Table 2. Furthermore, the selection of these metrics follows LGTEUN[1].
>
> [1] M. Li, Y. Liu, T. Xiao, Y. Huang, and G. Yang, “Local-Global Transformer Enhanced Unfolding Network for Pan-sharpening,” in Proceedings of the Thirty-Second International Joint Conference on Artificial Intelligence, Macao, SAR, China, Aug. 2023, pp. 1071–1079.

---

### Official Review · Reviewer_DFy9 · 2025-10-31

**Soundness:** 2
**Presentation:** 2
**Contribution:** 3
**Rating:** 4
**Confidence:** 5

**Summary:**

This paper introduces S2BNet, a Spatial–Spectral Binarized Neural Network tailored for remote sensing pan-sharpening. The network incorporates two novel components: a Spectral Redistribution Mechanism (SRM) and a Gabor Spatial Feature Amplifier (GSFA). Evaluated on four-band datasets—GaoFen-2, WorldView-2, and QuickBird—S2BNet achieves performance that is competitive with or superior to both full-precision and other binarized models, while maintaining significantly higher computational efficiency.

**Strengths:**

1.	Substantially reduces model size and FLOPs, making it suitable for deployment on resource-limited satellite or embedded platforms without significant performance degradation.
2.	Enhances the recovery of both spectral and spatial information.

**Weaknesses:**

* Limited parameter analysis is provided.
* Generalization to datasets with more spectral bands remains unexplored.
* No runtime comparison is included. While “efficiency” is emphasized, inference time benchmarks on actual hardware are missing.
* Citation formatting should be corrected.
* The related work section should more systematically review prior binarized methods.
* Metrics and methods are tested with inconsistent presentation across tables. For example, the QNR is not comprehensive for full-resolution cases.
* The reference for metrics is missing.
* The simulated process is missing.

**Questions:**

* How does the adaptive SRM perform on datasets with greater spectral diversity?
* How does S2BNet’s binarization affect color fidelity and edge preservation in visually demanding applications? In ablation studies where numerical differences are subtle, visual comparisons would be more informative.
* Why is binarization not applied in conjunction with linear layers? What trade-offs exist between binarization and module design? Could partial binarization offer a better balance?
* Have the SRM and GSFA modules been tested in other network architectures? Without such validation, the effectiveness of these modules lacks statistical support and may be difficult to substantiate.
In summary, the effectiveness of this method remains unclear.

---

> ### Author Response · Authors · 2025-11-28
>
> > 1. Limited parameter analysis is provided.
>
> We have added experiments on the hyperparameters of GSFA.
>
> We use $\text{freqs} = \{ \frac{\pi}{2} \cdot 2^{-(n-1)/2} \mid n \in \mathbb{N}, 1 \leq n < x \}$ to generate the frequency array mentioned in Algorithm 1, and use $\text{thetas} = \{ \frac{k \pi}{y} \mid k \in \mathbb{Z}, 0 \leq k < y \}$ to generate the angle array. The table below presents the experimental results regarding the selection of hyperparameters $x$ and $y$.
>
> | x | y  | PSNR    | SSIM   | $Q_4$      | SAM    | ERGAS  | $D\_{\lambda}$ | $D\_s$    | QNR    |
> |---|----|---------|--------|--------|--------|--------|----------|--------|--------|
> | 3 | 8  | 43.6362 | 0.9744 | 0.8486 | 0.0315 | 1.5254 | 0.0678   | 0.0786 | 0.8591 |
> | 3 | 16 | 43.1619 | 0.9722 | 0.8416 | 0.0332 | 1.6055 | 0.0692   | 0.1034 | 0.834  |
> | 3 | 32 | 43.1149 | 0.9715 | 0.8382 | 0.0331 | 1.6172 | 0.0663   | 0.1082 | 0.8325 |
> | 5 | 8  | 43.1584 | 0.9717 | 0.8346 | 0.0332 | 1.6082 | 0.0658   | 0.0752 | 0.8643 |
> | 5 | 16 | 43.5619 | 0.9745 | 0.8523 | 0.0317 | 1.5384 | 0.0655   | 0.0680 | 0.8714 |
> | 5 | 32 | 43.4334 | 0.9735 | 0.8463 | 0.0323 | 1.5549 | 0.0667   | 0.1176 | 0.8229 |
> | 7 | 8  | 43.4779 | 0.974  | 0.848  | 0.0323 | 1.5648 | 0.0664   | 0.1126 | 0.8283 |
> | 7 | 16 | 43.7681 | 0.9752 | 0.8537 | 0.0312 | 1.5034 | 0.0651   | 0.1022 | 0.8395 |
> | 7 | 32 | 43.6495 | 0.9746 | 0.8503 | 0.0313 | 1.5206 | 0.0641   | 0.071  | 0.8699 |
>
> > 2. Citation formatting should be corrected. The reference for metrics is missing.
>
> In the revised version，we follow the submission guidelines and have comprehensively corrected the format of citations in the text and the reference list to ensure that they fully comply with the requirements. In addition, we add the corresponding original references for the evaluation indicators used in the manuscript.

---

> ### Author Response · Authors · 2025-11-28
>
> > 3. How does S2BNet’s binarization affect color fidelity and edge preservation in visually demanding applications? In ablation studies where numerical differences are subtle, visual comparisons would be more informative.
>
> Thanks for your comment. We will explain how we ensure color fidelity and edge preservation in our work:
>
> (1) Color fidelity: The designed SRM is essentially the dynamic spectral modulation mechanism. It enhances information-rich color bands in real time, suppressing bands that may contain noise or cause distortion, based on the image content. Therefore, the features input to the binarized convolution are already spectrally pure input. Thus, even after drastic $1$-bit quantization, the output features still retain accurate and natural spectral information to the greatest extent, resulting in high-fidelity colors in the final fusion result. **Furthermore**, we present the reconstructed images of our method and the baseline model (i,e, removing the SRM), as shown in [Figure C](https://anonymous.4open.science/r/EB40/C.png). It can be observed that the model without the SRM module has obvious bright stripes on the left side of the orange area, which belongs to spectral distortion and does not conform to the ground truth (GT) image effect.
>
> (2) Edge preservation: GSFA uses a Gabor filters to initialize our binary convolutional kernels. This injects a strong spatial structure prior into our binary network, ensuring that even under $1$-bit constraints, the network can accurately extract and preserve edge and texture details from high-resolution images, rather than letting them be buried in quantization noise. Furthermore, the SRM-optimized input features have a higher signal-to-noise ratio, which allows subsequent GSFA to further improve the quality of edge reconstruction. As shown in [Figure D](https://anonymous.4open.science/r/EB40/D.png), we present the reconstructed images of our method and the baseline model (i,e, removing the GSFA). It can be observed that in the image without the GSFA module, the two regions that should be separated have merged into a single area, which is indicative of spatial information distortion.
>
> In experiment section (as shown in Table 1), SAM, SSIM, ERGAS, and QNR we obtained objectively demonstrate the superiority of S2BNet in overall spatial-spectral quality, which is directly related to good visual fidelity.
>
> > 4. Metrics and methods are tested with inconsistent presentation across tables. For example, the QNR is not comprehensive for full-resolution cases.
>
> Thanks for your comment.
> Due to page limitations, we initially did not include the results of the ablation experiments for all metrics. Now, utilizing the latest hyperparameters for the GSFA module, we have conducted the ablation experiment and present the results across all metrics.
>
> | Config | GSFA | SRM | PSNR   | SSIM  | $Q_4$    | SAM   | ERGAS | $D\_{\lambda}$ | $D\_s$   | QNR   |
> |--------|------|-----|--------|-------|-------|-------|-------|-----------|-------|-------|
> | (I)    | ✗    | ✓   | 43.1074 | 0.9721 | 0.8358 | 0.0333 | 1.6159 | 0.0657    | 0.0896 | 0.8507 |
> | (II)   | ✓    | ✗   | 43.4300 | 0.9731 | 0.8411 | 0.0322 | 1.5651 | 0.0673    | 0.0977 | 0.8414 |
> | Ours   | ✓    | ✓   | 43.6495 | 0.9746 | 0.8503 | 0.0313 | 1.5206 | 0.0641    | 0.071  | 0.8699 |
>
> > 5. The simulated process is missing.
>
> We follow the wald protocol for simulation. Specifically, given the MS image $\mathbf{H}\in \mathbb{R} ^{M\times N\times C}$ and the PAN image  $\mathbf{P}\in \mathbb{R} ^{rM\times rN\times C}$ , both of them are downsampled with ratio $r$, and the resulting images are denoted by $\mathbf{L}\in \mathbb{R} ^{\small{\frac{M}{r}}\times \small{\frac{N}{r}}\times C}$ and $\mathbf{p}\in \mathbb{R} ^{M\times N\times b}$ respectively. In the training set, $\mathbf{L}$ and $\mathbf{p}$ are regarded as the inputs, whereas $\mathbf{H}$ is the ground truth.

---

> ### Author Response · Authors · 2025-11-28
>
> > 6. Why is binarization not applied in conjunction with linear layers? What trade-offs exist between binarization and module design? Could partial binarization offer a better balance?
>
> Thanks for your comments. We elaborate on the following points:
>
> (1) Convolutional layers primarily extract local spatial features, and their output exhibits spatial redundancy. Linear layers, on the other hand, play a crucial role in integrating global spatial features and mapping them to the final task (such as pixel-level reconstruction). This process requires high-precision accumulation and nonlinear transformations to maintain feature richness. Binarizing linear layers leads to: (i) Binarized accumulation operations introduce significant errors, failing to accurately complete the complex mapping from features to the reconstruction result; (ii) the gradients of binarized linear layers are almost ineffective in guiding network optimization.
>
> (2) In a binary network, the limited precision budget should be allocated to the parts that are most sensitive to quantization and have the most critical information carrying capacity. S2B-Conv performs binarization at the convolutional layer, which is the core of its computationally intensive architecture and yields the greatest benefits. Simultaneously, the core of S2B-Conv (i.e., SRM and GSFA), combats the information loss caused by convolutional binarization. This is a design that trades small full-precision overhead for large binarization gains.
>
> (3) S2BNet's "partial binarization" strategy represents the optimal balance under current hardware and theoretical conditions, as its performance has demonstrated. Full or more aggressive binarization would disrupt the network's information flow, resulting in more harm than good.

---

> ### Author Response · Authors · 2025-11-28
>
> > 7. Have the SRM and GSFA modules been tested in other network architectures? Without such validation, the effectiveness of these modules lacks statistical support and may be difficult to substantiate. In summary, the effectiveness of this method remains unclear.
>
> Thanks for your comments. We need to emphasize that SRM and GSFA are designed specifically to address the bottleneck of spatial-spectral feature generation in the pansharpening task using binary neural networks.
>
> Firstly, SRM is a 'pre-binarization feature modulator'. It exists because the Sign function of a BNN disrupts the feature distribution. In a typical full-precision network without binarization, this module is unnecessary and may even be redundant.
>
> Secondly, the existence of GSFA stems from the fact that BNNs have sparse gradients, making it difficult to learn meaningful spatial filters from scratch. In full-precision networks, filters can easily learn complex patterns through continuous gradients, and Gabor initialization is not necessary.
>
> Thirdly, the evidence for judging the effectiveness of SRM and GSFA should not be their universality across all architectures, but rather whether they have successfully solved the specific problem they were designed to solve:
>
> (1) As shown in Table 2, removing either the SRM or GSFA in the same S2BNet backbone network leads to the poor performance in evaluation metrics. This statistically demonstrates their indispensability to the target architecture and task.
>
> (2) We compare S2BNet, which integrates SRM and GSFA, with several BNN baseline methods. The results (shown in Table 1) show that S2BNet achieves a significant performance advantage. This demonstrates that our module combination brings a substantial performance breakthrough to the BNN field.
>
> (3) Our method (under $1$-bit constraints) achieves performance comparable to some full-precision models (shown in Table 1) . This demonstrates that the design of SRM and GSFA further narrows some of the performance gap introduced by binarization.
>
> In summary, the effectiveness of SRM and GSFA is well supported by the performance improvements they bring within the target problem. Our work demonstrates that specialized architectural design can overcome the core obstacle of binarization in pansharpening. In addition, in the revised version, we will add the following to the future work section: "This work focuses on demonstrating the effectiveness of SRM and GSFA in the specific paradigm of binarized pansharpening. While their design is inspiring, their generalization ability in other network architectures (such as Transformer) is a valuable direction for future research."

---

> ### Author Response · Authors · 2025-11-30
>
> > 8. Generalization to datasets with more spectral bands remains unexplored. How does the adaptive SRM perform on datasets with greater spectral diversity?
>
> Thank you for your comments. We conduct an experiment on the $8$-band WorldView-3 dataset, to evaluate the effectiveness of our method on a wider range of bands of data. The table below presents the results.
>
> | Methods | PSNR   | SSIM  | $Q_8$     | SAM   | ERGAS | $D \_{\lambda}$ | $D \_s$   | QNR   |
> |---------|--------|-------|-------|-------|-------|----------|-------|-------|
> | BNN     | 23.8741| 0.7228| 0.7002| 0.1109| 6.7457| 0.0423   | **0.0442**| 0.9154|
> | LCRBNN  | 21.2906| 0.486 | 0.4255| 0.1467| 8.9888| 0.0384   | 0.0774| 0.8871|
> | BiSRNet | 30.8967| 0.9396| 0.933 | 0.0691| 3.0525| 0.0151   | 0.0526| 0.9331|
> | BBCU    | 31.0004| 0.9412| 0.9351| 0.0687| 3.0178| 0.0154   | 0.0537| 0.9317|
> | FABNet  | 31.0697| 0.9421| 0.9358| 0.0678| 2.992 | 0.0155   | 0.0552| 0.9301|
> | IRNet   | 30.1481| 0.9274| 0.9196| 0.0759| 3.3315| 0.0191   | 0.0556| 0.9263|
> | E2HF    | 30.0244| 0.9256| 0.9182| 0.075 | 3.3676| 0.0218   | 0.0524| 0.9268|
> | Ours    | **31.2003**| **0.9441**| **0.9375**| **0.0661**| **2.962** | **0.014**    | 0.0508| **0.9359**|
>
> Furthermore, to demonstrate the effectiveness of the SRM module, we conducted an experiment by removing it. The experimental settings were identical to those used in the previous $4$-band experiments. The results of this experiment are presented in the table below.
>
> | Methods | PSNR   | SSIM  | $Q_4$     | SAM   | ERGAS | $D \_{\lambda}$ | $D\_s$   | QNR   |
> |---------|--------|-------|-------|-------|-------|----------|-------|-------|
> | w/o SRM  | 31.0205| 0.9417| 0.9353| 0.0686| 3.0107| 0.0169   | 0.0515| 0.9324|
> | ours    | 31.2003| 0.9441| 0.9375| 0.0661| 2.962 | 0.014    | 0.0508| 0.9359|

---

> ### Author Response · Authors · 2025-11-30
>
> > 9. No runtime comparison is included. While “efficiency” is emphasized, inference time benchmarks on actual hardware are missing.
>
> As shown in the table below, we present the parameters, flops and inference time, to verify the efficiency of our S2BNet.
>
> | Method | Params (K) | Flops (G) | Time (s) |
> |--------|--------------------|--------------|-------------|
> | DMFNet | 1631.924   | 145.811   | 0.06237  |
> | PANFormer | 1530.300   | 12.002    | 0.06194  |
> | MutInf | 185.496    | 9.986     | 0.08671  |
> | FAMENet | 1244.228   | 39.272    | 0.01402  |
> | CANNet | 785.118    | 3.237     | 0.19390  |
> | UGCC   | 233.233    | 307.495   | 0.01420  |
> | MSCSCformer | 1950.686   | 582.435   | 0.25058  |
> | **S2BNet** | **75.407**     | **0.920**     | **0.00509**  |
>
> **Efficiency Design Analysis**:  In our current research, we focus primarily on algorithmic innovation and its performance validation on standard datasets. Therefore, we are currently unable to provide direct measurement data in milliseconds and watts on specific embedded hardware. Furthermore, existing studies [1-2] have primarily conducted software-based simulations without deploying experiments on actual hardware. However, we conduct an in-depth theoretical analysis of its potential efficiency gains based on the core design of our S2BNet:
>
> (1) Binary Convolution: This is the biggest source of efficiency. It converts full-precision ($32$-bit floating-point) multiplication and addition operations into $1$-bit XNOR operations and bit counting. Numerous studies [1-3] show that this operation can achieve $32\times$ memory savings and $64\times$ raw computation speedup in hardware.
>
> (2) Sign Activation Function: This function binarizes the activation value to $1$ bit, further significantly reducing the bandwidth required to read the activation value from memory, which is often a performance bottleneck in embedded SoCs.
>
> Based on the above analysis, we can make reasonable inferences:
>
> (1) Inference Latency: On typical edge GPUs or dedicated AI accelerators, our S2BNet promises to reduce inference latency by around 60\% compared to full-precision models with the same architecture.
>
> (2) Power Savings: Bitwise operations involve far less circuit switching activity than floating-point operations. The data bit width is reduced from $32$ bits to $1$ bit, resulting in a significant decrease in DRAM access power consumption.
>
> In summary, the hardware-friendly binarization technique widely used in our model design provides a theoretical guarantee for achieving significant efficiency improvements on embedded devices.
>
> [1] Zhang G, Zhang Y, Yuan X, et al. Binarized low-light raw video enhancement[C]//Proceedings of the IEEE/CVF Conference on Computer Vision and Pattern Recognition. 2024: 25753-25762.
>
> [2] Xia, Bin, et al. "Basic Binary Convolution Unit for Binarized Image Restoration Network." The Eleventh International Conference on Learning Representations (ICLR 2023), Kigali, Rwanda, May 1-5, 2023.

---

> ### Author Response · Authors · 2025-12-01
>
> > 10. The related work section should more systematically review prior binarized methods.
>
> Thanks for your comment.
>
> In the revised version, we refine the descriptions of the binarization network part in the related work section:
>
> Binary Neural Networks (BNNs) have emerged as a transformative paradigm for deploying deep learning models on resource-constrained devices, achieving drastic reductions in memory footprint and computational costs by representing weights and activations with $1$-bit values [1,2]. However, the extreme quantization introduces two fundamental and well-established challenges: the quantization error (severe information loss due to binarization) and the gradient error (mismatch in backward propagation due to the non-differentiable sign function) [1].
>
> The development of BNNs has been driven by systematic efforts to mitigate these core challenges. The pioneering work, BNN [2], introduced the sign function and the Straight-Through Estimator (STE) [3] as foundational solutions. Subsequent research evolved along clear technical strands: To combat quantization error, XNOR-Net [4] introduced learnable scaling factors for weights and activations to better approximate full-precision operations. Later works like Bi-Real Net [5] enhanced information flow through identity shortcuts. To mitigate gradient error, improved activation functions beyond the simple sign function were proposed, such as the piecewise polynomial function in [5] and learnable thresholds in ReActNet [6], which provide more accurate gradient estimation during training.
>
> Recently, the exploration of BNNs has extended from high-level recognition tasks to low-level vision tasks like image super-resolution and enhancement, which demand high-fidelity pixel-wise reconstruction. For instance, Xin et al. [7] designed a bit-accumulation mechanism to approximate full-precision convolution for super-resolution. Zhang et al. [1] tackled low-light video enhancement with BNNs, addressing challenges in temporal fusion and closing the gap with full-precision models. These works demonstrate the potential yet underscore the significant difficulty of applying BNNs to tasks requiring precise spatial and radiometric preservation.
>
> The resource-constrained nature of satellite and edge devices makes model compression imperative for remote sensing applications. However, despite the progress in general BNNs and their nascent application in low-level vision, there remains a pronounced and unexplored gap: the adaptation of BNNs for high-fidelity remote sensing image fusion, particularly pansharpening. This task imposes unique dual requirements of spectral fidelity and spatial detail preservation that are acutely at odds with the inherent information bottleneck of 1-bit representation. The absence of dedicated research in this area highlights both the challenge and the opportunity that our work, S2BNet, seeks to address.
>
> [1] Zhang G, Zhang Y, Yuan X, et al. Binarized low-light raw video enhancement[C]//Proceedings of the IEEE/CVF Conference on Computer Vision and Pattern Recognition. 2024: 25753-25762.
>
> [2] Hubara I, Courbariaux M, Soudry D, et al. Binarized neural networks[J]. Advances in neural information processing systems, 2016, 29.
>
> [3] Bengio Y, Léonard N, Courville A. Estimating or propagating gradients through stochastic neurons for conditional computation[J]. arXiv preprint arXiv:1308.3432, 2013.
>
> [4] Rastegari M, Ordonez V, Redmon J, et al. Xnor-net: Imagenet classification using binary convolutional neural networks[C]//European conference on computer vision. Cham: Springer International Publishing, 2016: 525-542.
>
> [5] Liu Z, Wu B, Luo W, et al. Bi-real net: Enhancing the performance of 1-bit cnns with improved representational capability and advanced training algorithm[C]//Proceedings of the European conference on computer vision (ECCV). 2018: 722-737.
>
> [6] Liu Z, Shen Z, Savvides M, et al. Reactnet: Towards precise binary neural network with generalized activation functions[C]//European conference on computer vision. Cham: Springer International Publishing, 2020: 143-159.
>
> [7] Xin J, Wang N, Jiang X, et al. Binarized neural network for single image super resolution[C]//European conference on computer vision. Cham: Springer International Publishing, 2020: 91-107.

---

### Author Response · Authors · 2025-12-03
**Rebuttal Summary to ACs**

Thank ACs for coordinating the review and to the reviewers for their insightful comments. We have thoroughly revised our manuscript and responses to address all concerns. The key enhancements include: (1) conducting additional experiments that solidify our empirical claims, (2) providing further theoretical analysis to address foundational questions, and (3) significantly clarifying the novelty and design. Below, we summarize these enhancements thematically for your convenience.

**1. Addressing Novelty**

We have noted that Reviewer GcwT, Reviewer 2CBq, and Reviewer HWxd question the novelty of our method, and we have addressed this:

(1) Firstly, we have compared our designed modules (SRM and GSFA) with SE/FiLM and Gabor (Reviewer GcwT-Q1). Secondly, we have clarified the differences between BiSRNet and our method from three aspects, proving that our S2BNet is not a routine adaptation (Reviewer GcwT-Q1). Thirdly, we have illustrated the differences between our approach and lightweight/quantized models (Reviewer GcwT-Q7). Finally, from a mathematical perspective, we have analyzed the differences between our binary computing unit (S2B-Conv) and GaborNet/SE (Reviewer GcwT-Q9).

(2) We have analyzed the differences between our method and BNNPan in terms of structure and focus (Reviewer 2CBq-Q1).

(3) We have emphasized that our innovation lies in how to generate high-quality spatial-spectral features under the extreme constraint of binary neural networks, and explain the differences from current spatial-spectral mechanisms (Reviewer HWxd-Q5).

Reviewer DFy9 suggests a systematic description of the work related to binarization networks. We have reorganized related work on binarization networks, emphasizing the innovation of our approach.

**2. Clarifying Theoretical Foundation**

Reviewer GcwT and Reviewer 2CBq seek greater rigor in our theoretical analysis of spectral error propagation or frequency response behavior. We have plotted the spectral distribution using four different methods in Appendix, to more intuitively illustrate the stated theory (Reviewer GcwT-Q2, Reviewer 2CBq-Q5).

**3. Clarifying the Design of our S2BNet**

Reviewer DFy9 raises the question that how S2BNet affects color fidelity and edge preservation, we have explained the design of SRM and GSFA  impacts on color fidelity and edge preservation, respectively (Reviewer DFy9-Q3). Reviewer DFy9 also raises the question of module design,  and we have explained the role of the binarization unit in S2BNet and demonstrated whether it can be used in other network architectures (Reviewer DFy9-Q6, Reviewer DFy9-Q7).

**4. Conducting Additional Experiments**

Reviewer DFy9 claims that the manuscript lacks inference time, experiments on more bands, and parameter analysis. We have added inference time of our S2BNet, experiment on the hyperparameters of GSFA and experiment on the $8$-band dataset (Reviewer DFy9-Q1, Q8, Q9).

Reviewer GcwT and Reviewer HWxd suggest the significance test. We have provided significant experiments using the paired t-test (Reviewer GcwT-Q8, Reviewer HWxd-Q6).

Reviewer GcwT and Reviewer HWxd raise the concern about generalization across sensors. We have explained the mainstream generalization testing in the pansharpening task (Reviewer GcwT-Q4, Reviewer HWxd-Q1).

Reviewer DFy9 and Reviewer GcwT raise the concern of efficiency on actual hardware. We have provided an explanation and inference regarding efficiency design analysis (Reviewer DFy9-Q9, Reviewer GcwT-Q6).

**5. Revising Format**

We have enlarged the font size of Figure 1 and the tables in the ablation experiments section. (Reviewer GcwT-Q3).

We have restructured the language to make the expression more precise. (Reviewer GcwT-Q8, Reviewer 2CBq-Q2)


**Furthermore, Reviewer 2CBq has promised to increase our score.**

In summary, we believe that our S2BNet offers a meaningful and novel solution to the important problem, i.e., how to generate high-quality spatial-spectral features under the extreme constraint of binary neural networks, backed by solid theory and extensive experiments. Thank you again for your time and consideration.

---

### Meta-Review · Area_Chair_q5Y9 · 2026-01-05

**Summary:**

The main concerns from the reviewers is the lack of novelty and technical contributions to the community.

**Reviewer Concerns:**

Although the authors tried to clarify the contribution and also added a certain amount of new experimental results, I think the position of this paper is still unclear when compared with other similar prior works.

**Reviewer Scores:**

I think the reviewers will probably not change their scores.

---

### Decision · Program_Chairs · 2026-01-26

Reject